

# Geometric amplification and suppression of ice-shelf basal melt in West Antarctica

Jan De Rydt[1] and Kaitlin Naughten[2]

[1]Department of Geography and Environmental Sciences, Northumbria University, Newcastle upon Tyne, UK
[2]British Antarctic Survey, Cambridge, UK

**Correspondence:** Jan De Rydt (jan.rydt@northumbria.ac.uk)

**Abstract.** Glaciers along the Amundsen Sea coastline in West Antarctica are dynamically adjusting to a change in ice-shelf mass balance that has triggered their retreat and speed-up prior to the satellite era. In recent decades, the ice shelves have continued to thin, albeit at a decelerating rate, whilst ice discharge across the grounding lines has been observed to increase by up to 100% since the early 1990s. Here, the ongoing evolution of ice-shelf mass balance components is assessed in a high-resolution coupled ice-ocean model that includes the Pine Island, Thwaites, Crosson and Dotson ice shelves. For a range of idealized ocean-forcing scenarios, the combined evolution of ice-shelf geometry and basal melt rates is simulated over a 200-year period. For all ice-shelf cavities, a reconfiguration of the 3D ocean circulation in response to changes in cavity geometry is found to cause significant and sustained changes in basal melt rate, ranging from a 75% decrease up to a 75% increase near the grounding lines, irrespective of the far-field forcing. These previously unexplored feedbacks between changes in ice-shelf geometry, ocean circulation and basal melting have a demonstrable impact on the net ice-shelf mass balance, including grounding line discharge, at multidecadal timescales. They should be considered in future projections of Antarctic mass loss, alongside changes in ice-shelf melt due to anthropogenic trends in the ocean temperature and salinity.

## 1 Introduction

Widespread dynamic thinning of glaciers along the Amundsen coastline in West Antarctica has contributed about 2000 giga-tonnes of ice, or about 5.5 mm, to global mean sea level rise between 1979 and 2017 (Rignot et al., 2019; Shepherd et al., 2019). The region includes the Pine Island, Thwaites, Pope and Smith glaciers (Fig. 1), which currently account for approximately 45% of excess ice discharge from the Antarctic Ice Sheet. The timing and underlying cause for the concurrent and spatially co-herent imbalance of these glacier basins remains a subject of ongoing research (Holland et al., 2022). However, it is understood that a critical role in the present-day evolution of the region's mass balance is played by ocean-induced ablation of its floating ice shelves (Pritchard et al., 2012; Gudmundsson et al., 2019; Smith et al., 2020). In future decades to centuries, numerical mass balance projections indicate that the Amundsen basin is likely to remain Antarctica's dominant contributor to sea level



rise (Seroussi et al., 2020; Edwards et al., 2021), despite significant uncertainties in climate forcing and poorly represented physical processes such as ice-shelf melting and temporal changes in ice rheology, basal sliding and ice front location.

A known role in the future evolution of the Ice Sheet's mass balance is played by the evolving buttressing capacity of the ice shelves. Any differences in back pressure due to changes in the structural integrity or geometry of the ice shelves, affect the force balance at the grounding line, and have the potential to impact on the dynamics of the upstream grounded ice (see e.g. Haseloff and Sergienko (2018) and Pegler (2018)). For this reason, changes in the mass balance of the ice shelves play a crucial role in moderating future mass loss from the Antarctic Ice Sheet. At present, ice shelves in the Amundsen Embayment

have a negative net mass balance (Adusumilli et al., 2018) as losses from basal ablation and ice-front calving outweigh the input of mass through grounding line fluxes and, to a much smaller extent, surface accumulation. This imbalance is the direct cause for the dynamic response of the adjacent glaciers (Gudmundsson et al., 2019).

Whilst the link between contemporary ice-shelf thinning and increased grounding line flux of Pine Island, Thwaites and other glaciers in the region has been well established, the future mass balance of the ice shelves and related impacts on grounding

line discharge remain uncertain. In recent decades, ice-shelf thinning rates have decreased (Adusumilli et al., 2018) due to an up to two-fold increase in grounding line flux (Mouginot et al., 2014; Davison et al., 2023; Otosaka et al., 2023) and in the absence of a demonstrable positive anomaly in the region's ocean heat content that could sustain anomalously high melt rates (Jenkins, 2016). A continued decrease in ice-shelf thinning could drive the system towards a new steady state, although there is no evidence from numerical simulations that such a steady state can be obtained under present-day ocean and atmospheric

conditions (Arthern and Williams, 2017; Reese et al., 2023, e.g.).

The rate at which glaciers in the Amundsen basin will continue to lose mass over the next decades to centuries, is controlled by several interrelated mechanisms. Firstly, changes in ice front location, loss of pinning points and ice damage evolution have all been shown to strongly impact on ice-shelf buttressing and grounding line discharge (Lhermitte et al., 2020; Joughin et al., 2019, 2021; De Rydt et al., 2021). Secondly, climatic trends in ocean and atmospheric forcing can lead to a sustained shift

in ice-shelf basal and/or surface ablation, and promote long-term changes in ice-shelf mass balance (Jourdain et al., 2022b). A third, as-of-yet poorly understood process, is the potential feedback between changes in the geometry of ice-shelf cavities, and the ocean dynamics that determines the melt rates for a given far-field distribution of temperature and salinity. Whilst such melt-geometry feedbacks have been studied for idealized geometries (De Rydt and Gudmundsson, 2016, e.g.) and the Thwaites Ice Shelf (Holland et al., 2023), little is known about their possible long-term impact on the net mass balance of

the West Antarctic ice shelves. Nor has the impact of geometry-driven changes in melt been compared to natural (seasonal to decadal) variability or projected anthropogenic trends in far-field ocean conditions.

From an ocean modelling point of view, the past evolution of basal melt rates in the Amundsen Sea (Naughten et al., 2022) and future projections under varying climate change scenarios (Jourdain et al., 2022a), have been assessed for fixed, present-day cavity geometries. For the fast retreating glaciers in the Amundsen Sea, this assumption represents an important

limitation of current basal mass balance projections. In ice-sheet models, melt rates and their dependency on ice-shelf geometry are commonly simulated using simplified parameterizations that do not retain knowledge about the complex, evolving 3D distribution of heat and salt within the ice shelf cavities. State-of-the-art projections of future mass loss from the Antarctic





Ice Sheet, as presented by (Edwards et al., 2021, e.g.), therefore lack the physical basis to accurately represent the coupling between changes in cavity geometry and ice-shelf basal ablation.

The current work presents a step towards bridging the gap between both incomplete modelling approaches, and assesses the importance of mutual feedbacks between ocean-driven melt rates and changes in cavity geometry. Starting from a numerical representation of the present-day state of the Amundsen Embayment and adjacent Ice Sheet (Fig. 1), a series of 200-yr long simulations of the Pine Island, Thwaites, Crosson and Dotson glaciers and ice shelves is presented. All simulations were conducted using a newly developed configuration of the coupled ice sheet-ocean model Úa-MITgcm (Naughten and De Rydt,

2023), with a mutually evolving dynamical ice sheet and 3D ocean. Significant changes in the geometry of all ice-shelf cavities were simulated over the 200-yr period for a range of forcing scenarios, and a detailed analysis of the impact on the 3D cavity circulation and basal melt rates is presented. Geometrically-driven changes in melt are compared to changes caused by natural decadal variability in ocean conditions on the Amundsen continental shelf, and the significance of melt-geometry feedbacks for the overall mass balance of the ice shelves, including grounding line fluxes, is assessed.

The remainder of this paper is structured as follows. A detailed overview of the coupled ice-ocean model and experimental design is provided in Sect. 2. The necessary technical background for the analysis of ice-shelf basal melt rates in response to changes in cavity geometry is introduced in Sect. 3. This includes the definition of the 'cavity transfer coefficients' that link average melt rates in the vicinity of the grounding line to the thermal driving of the inflow across the ice front. In Sects. 4 and 5 the analysis of three numerical experiments is presented, focused on the ocean processes that drive changes in basal

melt in response to evolving ice-shelf cavities and the relative importance of decadal variations in far-field ocean conditions. A glaciological perspective and the importance of melt-geometry feedbacks for the net mass balance of the ice shelves, including grounding line discharge, is presented in Sect. 6, followed by conclusions and future perspectives in Sect. 7.

## 2   Numerical setup and experiments

Basal ablation of ice shelves in the Amundsen Sea locally exceeds $100\,\mathrm{m\,yr^{-1}}$ due to the presence of modified Circumpolar

Deep Water (mCDW) with temperatures several degrees above the local freezing point (Jacobs and Hellmer, 1996). Whilst mCDW is continuously present on the continental shelf (see Fig. 1), mean melt rates are strongly modulated by decadal variations in the thickness of the mCDW layer (Dutrieux et al., 2014). However, direct observations of ocean variability on the continental shelf remain sparse, and knowledge about the distribution of heat and salt within the ice-shelf cavities largely relies on the use of general circulation models.

The ocean currents that cross the ice front and transport thermal energy towards the ice-ocean interface, are steered by the topography of the seafloor and ice-shelf base. Away from sources of barotropic potential vorticity, the depth-averaged flow is approximately aligned with contours of constant water column thickness (Patmore et al., 2019, e.g.) and as a result, sharp gradients in topography such as the ice front (Bradley et al., 2022) or bathymetric sills (De Rydt and Gudmundsson, 2016; Zhao et al., 2019), can impose significant barriers to the flow. The baroclinic transport, on the other hand, is primarily driven

by melt-induced stratification, which is affected by geometrical factors such as the gradient of the ice-shelf base.





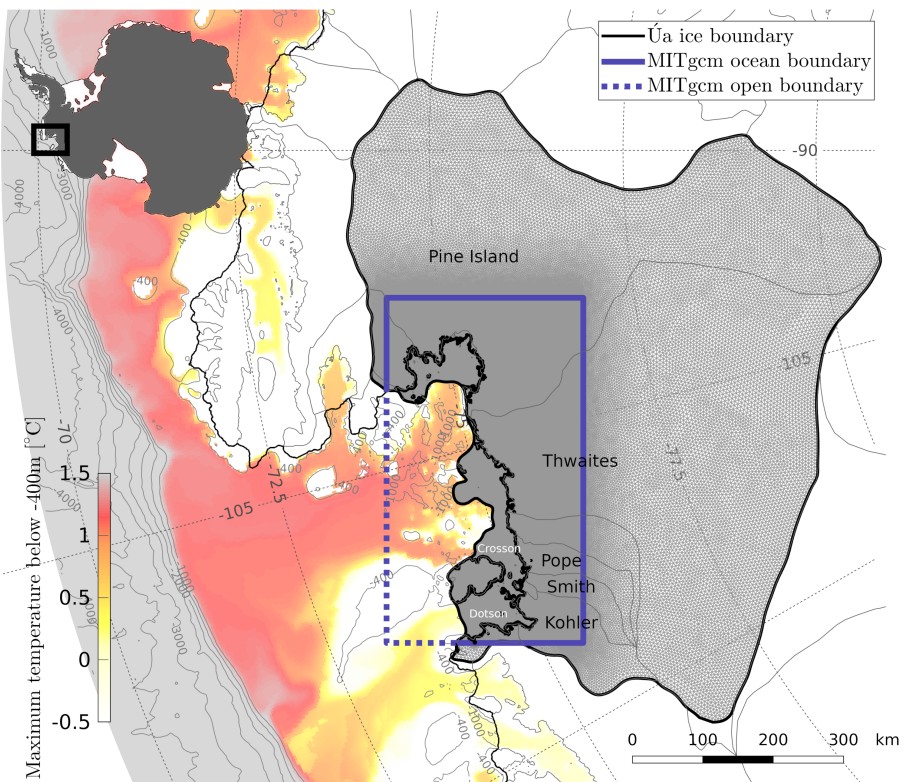

**Figure 1.** Study area, showing its location in West Antarctica (inset in the top left) and boundaries of the Úa (ice) and MITgcm (ocean) model domains. The Úa domain includes the Pine Island, Thwaites, Crosson and Dotson ice shelves, delineated in black, and drainage basins of their tributary glaciers (Rignot et al., 2019), shown in light grey. The dashed blue line corresponds to the open boundary of MITgcm, where ocean restoring conditions are imposed, as described in Sect. 2.1. The background colour scale corresponds to a 1994-2013 climatology of maximum ocean temperatures below 400 m, averaged over the ensemble of PACE simulations from Naughten et al. (2022). Bathymetric contour lines for the open ocean are from (Morlighem et al., 2020).

To simulate the 3D structure of ocean currents in the geometrically complex cavities of the Amundsen Sea, the MIT general circulation model (MITgcm) with thermodynamically active ice shelves (Marshall et al., 1997; Losch, 2008) was used. To capture the 2-way feedbacks between changes in ice-shelf geometry and basal melt whilst the ice sheet evolves over time, MITgcm was coupled to the ice flow model Úa (Gudmundsson et al., 2012; Gudmundsson, 2020), as described in detail below.
The coupled ice-ocean model, Úa-MITgcm, was first developed for simulations of an idealized, Pine Island-like setup by De Rydt and Gudmundsson (2016), and subsequently adapted by Naughten et al. (2021) (see their methods section for details). The model setup is described in more detail in Sect. 2.1, followed by an overview of the experimental design in Sect. 2.2.



## 2.1 Coupled ice-ocean model setup

The MITgcm domain covers a 138,000 km$^2$ region of the Amundsen continental shelf between 97°W and 116°W, and 73°S
and 77°S, delineated by the blue box in Figure 1. The domain includes the present-day Pine Island, Thwaites, Crosson and
Dotsen ice shelf cavities with a buffer of grounded ice to allow upstream migration of the grounding line. MITgcm solves the
Boussinesq and hydrostatic form of the Navier Stokes equations on an Arakawa C grid in polar stereographic coordinates with
a uniform horizontal resolution of 1.3 km. The $z$-coordinate levels consist of 80 layers with 20 m resolution down to a depth
of 1600 m, and 10 vertical levels with 40 m resolution down to 2000 m. MITgcm uses partially filled cells with a minimum
thickness of 1 m to allow greater flexibility of the piecewise linear representation of the bathymetry and ice-shelf draft. At
the northern and western open boundaries of the domain (dashed blue line in Figure 1), restoring conditions for temperature,
salinity and velocities were applied across a 6.5 km wide sponge layer, with restoring timescales ranging from 3 hrs at the
boundary to 6 hrs in the interior. Further details about the boundary conditions are provided in Sect. 2.2.

In all experiments ocean surface fluxes (heat, salt, momentum) were ignored, and variability in water mass properties was
imposed solely through restoring at the northern and western open boundaries. The absence of surface fluxes prevents a realistic
representation of the ocean mixed layer and eliminates polynya activity that can cause deep convection. However, for the
purpose of this study, such processes are thought to be of secondary importance. Rather than analyze the connection between
offshore atmospheric conditions, variability in water mass properties and basal melt, the focus will be on the links between
changes in cavity geometry and basal melt for given far-field ocean conditions. Moreover, the forthcoming analysis will be
primarily restricted to the analysis of thermohaline properties in the deepest parts of the cavities, which are thought to be
largely unaffected by changes in surface waters.

Ice-shelf melt rates in MITgcm were computed using the 3-equation formulation for conservation of heat, salt and mo-
mentum (Holland and Jenkins, 1999) with ambient temperature and salinity averaged across a 20 m layer below the ice shelf
(Losch, 2008) and a linear dependency of the exchange coefficients on the friction velocity averaged over the same layer.

The Úa domain includes the Pine Island, Thwaites, Dotson and Crosson ice shelves and their respective drainage basins
(Rignot et al., 2019), as shown by the black outline in Figure 1. The ice front was traced from a Landsat 5 image from 1997 and
kept fixed in all simulations. Úa uses finite element methods to solve the vertically integrated shallow shelf (SSA or SSTREAM)
equations of ice flow on an unstructured grid with linear triangles. The mean nodal spacing of the mesh is 1.5 km, with local
refinement down to 400 m in areas with high strain rates and strain rate gradients. In line with published recommendations
to minimize resolution-dependency of the grounding line dynamics (see e.g. Pattyn et al. (2012) and Cornford et al. (2016)),
adaptive mesh refinement down to 400 m was used within a 2 km buffer around the moving grounding line. In order to minimize
the need for interpolation between the MITgcm and Úa meshes, MITgcm nodes were included as a subset of the Úa nodes at
all times. The ice rheology was described using Glen's law with exponent $n = 3$, and basal sliding was parametrized using a
non-linear Weertman law with exponent $m = 3$. Ice viscosity and basal slipperiness, which are unknown fields in the rheology
and sliding parameterizations, were estimated using an inverse method, as described in App. A. Non-zero melt rates were only





applied to nodes of fully floating elements, following the conservative "no-melt parameterization" scheme in Seroussi and Morlighem (2018). A zero flow condition was imposed for the ice divide at the boundary of the mesh.

The Úa-MITgcm coupler (Naughten and De Rydt, 2023) controls the offline exchange of basal melt rates and ice-shelf geometry between the otherwise independent models. The coupling timestep, or frequency at which data is exchanged between models, was set to 30 days. At each coupling timestep, MITgcm melt rates averaged over the coupling period were linearly interpolated onto the Úa grid, and the new Úa ice draft was transferred to MITgcm. Both models were restarted from their final state at the end of the previous coupling timestep. The temperature and salinity of newly opened ocean cells were horizontally extrapolated from neighbouring cells, and velocities of the corresponding water column were corrected to preserve the barotropic transport and avoid a step-change in the flow divergence. In order to enable overturning in shallow water columns, a minimum of two vertical cells was achieved by (temporarily) lowering the bathymetry in MITgcm for small areas of the domain, in particular near the grounding line. Corrections to the bathymetry were less than the vertical resolution of the MITgcm grid.

A detailed overview of the input data sets, initialization and spin up of MITgcm, Úa and the coupled Úa-MITgcm configuration are provided in Appendix A.

## 2.2 Experimental design

The numerical experiments were designed with two key objectives in mind. Firstly, to quantify feedbacks between future changes in cavity geometry and basal melt rates, and secondly, to compare geometrically-driven changes in melt rate to variability caused by natural changes in ambient ocean conditions. All experiments were started after the model spin up, with an initial ice-sheet configuration that is close to the present-day state, as detailed in App. A.

In the first set of experiments, cavities were exposed to 200 years of time-invariant far-field ocean conditions, imposed as constant restoring of temperature, salinity and velocities at the open boundaries (Fig. 1). The absence of temporal variability in heat transport at the ocean boundaries allowed all changes in basal melt to be attributed to geometrical feedbacks, hence addressing the first objective outlined above. The restoring conditions were subsampled from an existing hindcast simulation (01/1997-12/2014) of the Amundsen Sea (Kimura et al., 2017), based on a MITgcm configuration with horizontal resolution of 0.1° in longitude and ERA-Interim atmospheric forcing.

The first experiment (see Table 1) corresponds to a 'high melt' scenario, henceforth referred to as *hi_melt*, with monthly average temperature, salinity and velocities from November 2002 imposed as restoring conditions at the open boundaries of the MITgcm domain. The corresponding ocean state has a shallow thermocline and is characteristic for high amplitude, low prevalence conditions within the 1997-2014 frequency distribution of simulated melt rates (see Figures 2 and A3).

The second experiment corresponds to an 'average melt' scenario, henceforth referred to as *av_melt*, with average oceanographic conditions from January 1998 imposed at the open boundaries. The corresponding ocean state is characteristic for average amplitude, high prevalence melt rates (Fig. 2), with the exception of the Crosson and Dotson cavities, where conditions are biased towards lower-than-average melt rates.




| Experiment | Boundary restoring | Ice-shelf geometry | Simulation time |
|---|---|---|---|
| *hi_melt* | montly average $T, S, U, V$ for 11/2002 | evolving | 200 yrs |
| *av_melt* | monthly average $T, S, U, V$ for 01/1998 | evolving | 200 yrs |
| *var_melt* | time-varying, 11 cycles of 01/1997 - 12/2014 | evolving | 198 yrs |
| *ref_melt* | time-varying, 11 cycles of 01/1997 - 12/2014 | fixed present-day | 198 yrs |

**Table 1.** Overview of the numerical experiments presented in the main part of the text. All boundary restoring conditions (temperature, salinity, velocities) were obtained from a regional MITgcm configuration of the Amundsen Sea (Kimura et al., 2017).

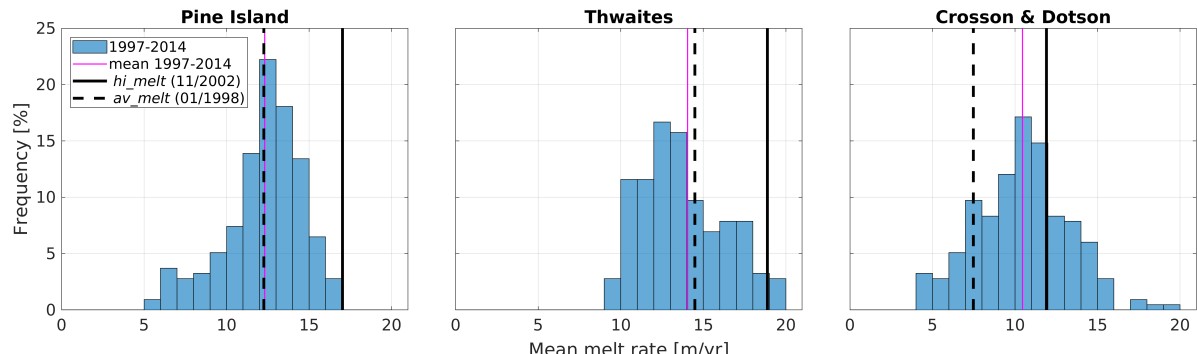

**Figure 2.** Frequency distribution of monthly mean melt rates for each ice shelf, as simulated during the first cycle of the *var_melt* experiment, covering the January 1997 to December 2014 period. The magenta line indicates the mean melt rate over the entire simulation period. The solid and dashed lines indicate melt rates that correspond to the ocean forcing applied in the *hi_melt* and *av_melt* experiments, respectively.

For completeness, a 'low melt' scenario with a deep thermocline was considered, which resulted in a persistently positive mass balance of the ice shelves, and a readvance of the grounding lines. Since this behaviour is unlikely to inform the future evolution of glaciers in the Amundsen basin, this scenario was not considered further.

In the second set of experiments, time-varying restoring conditions were applied at the MITgcm boundaries. The aim was not to carry out projections for different emissions scenarios, but rather to disentangle geometrical feedbacks from the sensitivity of melt to natural variability in the ocean state. Monthly averaged ocean conditions between January 1997 and December 2014 from (Kimura et al., 2017) were imposed at the MITgcm boundary, and linearly interpolated onto the model time. The 1997-2014 restoring conditions were cyclically repeated 11 times for a total of 198 simulation years. The frequency distribution of melt rates during the first cycle is shown in Figure 2. As well as a coupled ice-ocean simulation with a dynamic ice-shelf geometry (referred to as *var_melt*), a reference stand-alone ocean simulation with fixed ice-shelf cavities was carried out (referred to as *ref_melt*). For both experiments, the variability in basal ablation was analyzed within each 18-year forcing cycle, as well as between successive cycles.

Despite the idealized nature of the ocean forcing, experiments induce an ice-sheet response that is informative for sustained present-day ocean conditions (*av_melt* and *var_melt*) and future climate conditions dominated by warm periods (*hi_melt*). The





key advance of this work is the inclusion of ice-ocean feedbacks simulated by a 3D ocean model. As will be argued in Sect.
6, such feedbacks play an important role in the dynamical evolution of the Amundsen Sea glaciers, and results provide a first
step towards more definitive climate change scenario-based projections of mass loss in response to ocean-induced ice-shelf
thinning over the next decades to centuries.

## 3 Cavity transfer coefficients

In order to diagnose the feedbacks between changes in basal melt, imposed ocean boundary conditions and cavity geometry, a
number of 'cavity transfer coefficients' are introduced. Each time-varying coefficient will be shown to play a distinctive role
in linking the far-field ocean properties to the basal melt rates, while their values depend to greater or lesser extent on the
complex changes in cavity geometry. More precisely, spatially averaged basal melt rates can be expressed as a product of the
time-varying ocean thermal forcing at the ice front $T_{\star\mathrm{IF}}(t)$, the 'outer cavity transfer coefficient' $\mu(t)$, the 'thermal transfer
coefficient' $\epsilon_T(t)$, and the 'momentum transfer coefficient' $\epsilon_U(t)$, as follows:

$$m(t) = m_0 \epsilon_T(t)\epsilon_U(t)\mu(t)^2 \left(T_{\star\mathrm{IF}}(t)\right)^2 , \tag{1}$$

where $m_0$ is a constant defined in Eq. 5. The expression in Eq. 1 can be derived from the fundamental relationships between
basal melt and properties of the oceanic mixed layer directly beneath the ice shelf, as follows.

Under the assumption that turbulent mixing of heat towards the ice-ocean interface is dominated by shear instabilities in the
current, basal melt rates can be parameterized as a function of the thermal driving in the mixed layer, denoted by $T_\star$, and the
friction velocity relative to the ice base, denoted by $U_\star$ (Jenkins and Bombosch, 1995):

$$m(\mathbf{x},t) = m_0\, T_\star(\mathbf{x},t)\, U_\star(\mathbf{x},t), \tag{2}$$

with

$$
\begin{aligned}
T_\star(\mathbf{x},t) &= T(\mathbf{x},t) - T_f(\mathbf{x},t), &\text{(3)}\\
U_\star(\mathbf{x},t) &= C_d^{1/2} U(\mathbf{x},t), &\text{(4)}\\
m_0 &= \frac{\Gamma_T}{(L_i - c_i T_{\star i})/c}, &\text{(5)}
\end{aligned}
$$

where $T_f(\mathbf{x},t)$ is the salinity-dependent freezing point at the depth of the ice base, and $T_{\star i} = T_f - T_i$ where $T_i$ is the temperature
of the ice. Values $C_d = 2.5\times10^{-3}$, $\Gamma_T = 0.02$, $L_i = 3.4\times10^5\,\mathrm{Jkg}^{-1}$, $c_i = 2.0\times10^3\,\mathrm{Jkg}^{-1\circ}\mathrm{C}^{-1}$, and $c = 4.0\times10^3\,\mathrm{Jkg}^{-1\circ}\mathrm{C}^{-1}$
were used for the quadratic drag coefficient, turbulent heat exchange coefficient, enthalpy of fusion of ice, specific heat capacity
of ice and specific heat capacity of water, respectively. In all MITgcm simulations, basal melt rates were calculated using the
parameterization in Eq. 2, with $T(\mathbf{x},t)$ and $U(\mathbf{x},t)$ calculated as a weighted average of cell properties adjacent to the ice-ocean
interface, as described in Sect. 2.1.

Properties of the mixed layer, and hence basal melt rates, are controlled by the thermal forcing of water masses that enter
the cavity. In particular, the temperature of the mixed layer is modulated by mixing of the ambient water with meltwater at





the local freezing temperature. As a result, the thermal driving of the mixed layer can be defined as a fraction of the average thermal driving of the inflow, $T_{\star \text{in}}$:

$$T_\star(\mathbf{x}, t) = \epsilon_T(\mathbf{x}, t) T_{\star \text{in}}(t), \tag{6}$$

with $0 < \epsilon_T < 1$ a dimensionless coefficient (Jenkins et al., 2018), which will henceforth be referred to as the 'thermal transfer coefficient'.

Based on inherent properties of the conservation for mass, momentum, heat and salt within the mixed layer, it has been argued that the friction velocity, $U_\star$, also scales approximately linearly with the thermal forcing (Holland et al., 2008). Analogous to Eq. 6, a scaling coefficient, henceforth denoted by $\epsilon_U$ and refered to as 'momentum transfer coefficient', is introduced to express this relationship:

$$U_\star(\mathbf{x}, t) = \epsilon_U(\mathbf{x}, t) T_{\star \text{in}}(t), \tag{7}$$

where $\epsilon_U$ has dimensions m s$^{-1}$°C$^{-1}$. Combining Eqs. 2, 6 and 7 then leads to

$$m(\mathbf{x}, t) = m_0 \epsilon_T(\mathbf{x}, t) \epsilon_U(\mathbf{x}, t) \left( T_{\star \text{in}}(t) \right)^2, \tag{8}$$

which expresses local melt rates as a quadratic function of the thermal forcing, and two transfer coefficients, $\epsilon_T$ and $\epsilon_U$, which link the far-field driving to properties of the mixed layer.

To facilitate the analysis of temporal changes in melt rates, Eq. 8 can be spatially averaged. One viable choice is to calculate average basal melt rates for each ice shelf, and define $T_{\star \text{in}}$ as the average thermal driving of the inflow across the corresponding ice front. One shortcoming of this simple diagnostic, however, is the loss of information about changes in melt that occur over small areas of glaciological importance, in particular in the vicinity of the grounding line. Since the steepest slopes of ice shelf draft in the Amundsen Sea are typically found near the grounding line, ice-shelf wide averages are dominated by melt rates in the relatively extensive, shallow areas above the thermocline. Since the focus of this study is on geometrical changes and basal melt feedbacks in the deep interior of the cavities, a more appropriate diagnostic will be introduced below.

First, all cavities are divided in an 'outer region' and a 'deep interior', as illustrated for the Pine Island Ice Shelf cavity in Fig. 3. The deep interior is defined as the part of the cavity with ice draft below -400 m. This bound approximates the shallowest depth at which mCDW is currently found on the Amundsen continental shelf (Dutrieux et al., 2014; Jenkins et al., 2018), and the deep interior can be thought of as ice-shelf regions that are directly exposed to (a modified form of) the warmest waters. Secondly, $T_{\star \text{in}}$ is defined as the average thermal driving of the inflow across a vertical section along the -400 m ice draft contour, henceforth denoted by $T_{\star \text{DI}}(t)$, where DI refers to Deep Interior. A schematic representation of $T_{\star \text{DI}}(t)$ is shown in Figure 3, and calculated as follows:

$$T_{\star \text{DI}}(t) = \frac{1}{L} \int\limits_{b(\mathbf{x}) = -400} \frac{1}{H(l)} \left( \int\limits_{B(l)}^{-400} T_\star(l, z, t) \theta(U_n) \, \mathrm{d}z \right) \mathrm{d}l, \tag{9}$$

with $L = \int_{b(\mathbf{x}) = -400} \mathrm{d}l$ the horizontal length of the section, $b$ and $B$ the ice draft and seafloor depth respectively, $H = b - B$ the water column thickness, $T_\star = T - T_f$ where the freezing temperature $T_f$ is calculated at a given depth $z$, and $\theta$ a Heaviside



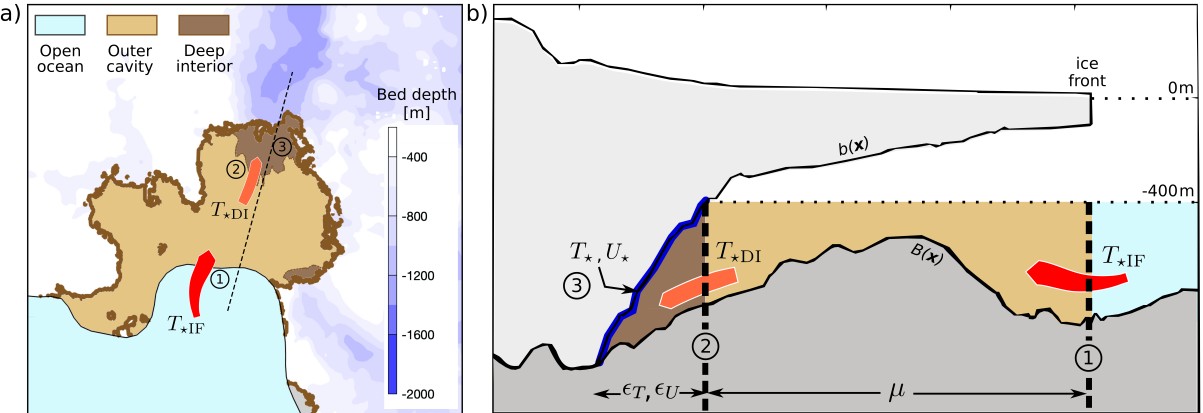

**Figure 3.** (a) Plan view and (b) section along the dashed line in a) of the Pine Island Ice Shelf, used to illustrate the concepts of the 'outer cavity' and 'deep interior', which correspond to regions of the ice-shelf cavity with draft above and below -400 m, respectively. Key locations referred to in the main text are numbered 1-3: 1) section at the ice front where thermal driving of the inflow below -400 m ($T_{\star\mathrm{IF}}$) is calculated, 2) section at the -400 m ice shelf draft contour where thermal driving of the inflow into the deep cavity is calculated ($T_{\star\mathrm{DI}}$), and 3) the oceanic mixed layer in the deep interior where the thermal deficiency ($T_\star$) and boundary current ($U_\star$) that determine the melt rates are calculated.

step function which is zero/one for negative/positive (i.e. outflowing/inflowing) normal velocities $U_n$. Finally, melt rates are

averaged over the deep interior and expressed as a function of $T_{\star\mathrm{DI}}$ using Eqs. 2, 6 and 7:

$$
\overline{m}_{\mathrm{DI}}(t) \coloneqq \frac{1}{\mathcal{A}} \int\limits_{b(\mathbf{x})\leq-400} m(\mathbf{x},t) \; \mathrm{d}\mathbf{x} = \frac{m_0}{\mathcal{A}} \int\limits_{b(\mathbf{x})\leq-400} T_\star(\mathbf{x},t)\,U_\star(\mathbf{x},t) \; \mathrm{d}\mathbf{x}
$$

$$
= m_0 \left( \frac{1}{\mathcal{A}} \int\limits_{b(\mathbf{x})\leq-400} \epsilon_T(\mathbf{x},t) \; \mathrm{d}\mathbf{x} \right) \left( \frac{1}{\mathcal{A}} \int\limits_{b(\mathbf{x})\leq-400} \epsilon_U(\mathbf{x},t) \; \mathrm{d}\mathbf{x} \right) \left( T_{\star\mathrm{DI}}(t) \right)^2
$$

$$
+ \; \mathrm{Cov}\left( T_\star, U_\star \right)
$$

$$
= m_0 \, \bar{\epsilon}_{T,\mathrm{DI}}(t)\, \bar{\epsilon}_{U,\mathrm{DI}}(t) \; \left( T_{\star\mathrm{DI}}(t) \right)^2 + \mathrm{Cov}\left( T_\star, U_\star \right), \tag{10}
$$

In the above derivation, spatial gradients of heat diffusion into the ice were ignored and a constant $T_{\star i} = T_f - T_i = -15°\mathrm{C}$ was assumed, such that $m_0 \approx 2.2 \times 10^{-4\,°}\,\mathrm{C}^{-1}$. Although spatial variations in $T_f$ were accounted for in the calculation of basal melt in MITgcm, corrections are small and will be ignored for the purpose of this analysis. Notations $\bar{\epsilon}_{T,\mathrm{DI}}$ and $\bar{\epsilon}_{U,\mathrm{DI}}$ were introduced to represent coefficients averaged over the deep interior of each ice shelf, and $\mathcal{A}(t) = \int_{b(\mathbf{x})\leq-400} I \; \mathrm{d}\mathbf{x}$ was used to denote the area of the ice shelf with draft below -400 m. It can be shown that the covariance term $\mathrm{Cov}\left(T_\star, U_\star\right)$ is typically an

order of magnitude smaller than the first term on the right hand side, and will be omitted from subsequent analysis.

Under certain conditions, such as simulations of idealized cavity geometries with a flat bathymetry, properties of the inflow at the ice front propagate into the deep cavity without significant modification. As a result, $T_{\star\mathrm{DI}}$ is approximately equal to the thermal driving of the inflow at the ice front, henceforth denoted by $T_{\star\mathrm{IF}}$ and schematically represented in Figure 3. For





realistic cavities with complex geometry and ocean dynamics, however, the ratio between $T_{\star\mathrm{DI}}$ and $T_{\star\mathrm{IF}}$ can be significantly
different from one. Indeed, even though the outer cavities in proximity of the ice front are flooded with mCDW, mixing of
the mCDW with subglacial meltwater and/or topographic blocking can prevent the inflowing water to reach the deep cavities
unmodified. As a result, the ocean properties that drive the onset of subglacial melt plumes in the vicinity of the grounding
line are not generally identical to properties of the inflow at the ice front. In order to characterize the relationship between
properties of the inflow at the ice front and the thermal driving in the deep interior, the dimensionless, time-dependent 'outer
cavity transfer coefficient' $\mu(t)$ is introduced:

$$T_{\star\mathrm{DI}}(t + \Delta t) = \mu(t)\,T_{\star\mathrm{IF}}(t)\,, \tag{11}$$

where $T_{\star\mathrm{IF}}$ is calculated as follows:

$$T_{\star\mathrm{IF}}(t) = \frac{1}{L}\int\limits_{\mathrm{ice\,front}} \frac{1}{H(l)}\left(\int\limits_{B(l)}^{-400} T_{\star}(l,z,t)\,\theta(U_n)\;\mathrm{d}z\right)\,\mathrm{d}l\,, \tag{12}$$

width $L = \int_{\mathrm{ice\,front}}\mathrm{d}l$ and $H(l) = -B(l) - 400$. The coefficient $\mu$ provides an aggregated measure for the impact of outer-
cavity ocean processes, including mixing and topographic steering, on the water properties that feed the buoyant melt water
plumes. The offset $\Delta t$ in Eq. 11 accounts for the time lag between water flowing into the cavity at the ice front and reaching the
deep interior, which is a measure of the cavity residence time. For a horizontal length scale of 100 km, which is characteristic
for the Amundsen Sea ice shelves, and an average barotropic circulation at the ice front between $0.01\,\mathrm{m\,s^{-1}}$ and $0.015\,\mathrm{m\,s^{-1}}$,
the residence time is on the order of a few months. Since the far-field ocean conditions vary predominantly at interannual to
decadal timescales, changes in the inflow evolve slowly compared to the cavity residence time, and $\Delta t$ can be set to zero to
good approximation.

Only currents below -400 m are considered in the calculation of $T_{\star\mathrm{IF}}$, for several reasons. Firstly, water masses that access
the cavity at depth form the most important source of basal meltwater production in the deep interior. Secondly, the cavity
inflow above -400 m predominantly alters the mixed layer properties and melt rates in the outer cavity. Although changes in
the ice-shelf draft at the ice front caused significant variations in the shallow currents in some of the Úa-MITgcm simulations,
they have little impact on the melt rates in the deep interior, and will not be analysed here. Finally, as pointed out in Sect. 2, the
model does not simulate surface processes in the open ocean, such as wind stress, or heat and salt fluxes. Properties of the upper
water column and deeper waters in areas with convective mixing are therefore not reliably represented. Several sensitivity tests
(not shown here) indicate that results presented in subsequent sections are not critically dependent on the choice of the cut-off
depth, and no notable differences were found if either -300 m or -500 m was used to define $T_{\star\mathrm{IF}}$ instead.

In summary, Eq. 10 and Eq. 11 can be combined to express the average melt of the deep interior cavity as a function of the
far-field thermal forcing, the thermal and momentum transfer coefficients $\epsilon_T$ and $\epsilon_U$, and the outer cavity transfer coefficient $\mu$,
as expressed in Eq. 1. To simplify the notation, the subscripts DI and overlines have been dropped from $\overline{m}_{\mathrm{DI}}$, $\overline{\epsilon_{T,\mathrm{DI}}}$ and $\overline{\epsilon_{U,\mathrm{DI}}}$.

Variations in average thermal driving of the inflow at the ice front ($T_{\star\mathrm{IF}}$) are strongly controlled by open-ocean dynamics
and climate drivers, whereas $(\epsilon_T, \epsilon_U, \mu)$ characterise the large-scale cavity dynamics, which is expected to depend on the cavity





geometry. The relative importance of $T_{\star\text{IF}}$ and the transfer coefficients in controlling the temporal evolution of the melt rates as the cavity geometry evolves, will be assessed next. To enable easy comparison between the different factors on the right hand side of Eq. 1, all quantities will be scaled by their respective values at $t = 0$:

$$\widetilde{m}(t) = m_0 \widetilde{\epsilon}_T(t) \widetilde{\epsilon}_U(t) \widetilde{\mu}(t)^2 \left( \widetilde{T}_{\star\text{IF}}(t) \right)^2, \tag{13}$$

where $\widetilde{m}(t) = \frac{m(t)}{m(t=0)}$, $\widetilde{\epsilon}_T(t) = \frac{\epsilon_T(t)}{\epsilon_T(t=0)}$, and so on.

## 4  Ice-ocean feedbacks

A logical first step in the diagnosis of geometry-melt feedbacks in the deep interior of the Amundsen Sea cavities, is to simulate the response of the coupled ice-ocean system for suppressed temporal variability of the inflow at the ice front, i.e. $\widetilde{T}_{\star\text{IF}}(t) \approx 1$. In doing so, the time-dependency of normalized basal melt rates ($\widetilde{m}(t)$ in Eq. 13) can be attributed solely to changes in the

transfer coefficients, which capture changes in the cavity geometry, and any contamination from far-field ocean variability is eliminated. The experiments *hi_melt* and *av_melt*, described in Sect. 2.2, aim to achieve this by imposing time-invariant restoring conditions at the MITgcm boundaries for 200 years.

At the beginning of the *hi_melt* experiment, average basal melt rates in the deep interior range from 22.4 m yr$^{-1}$ (Crosson) to 64.4 m yr$^{-1}$ (Pine Island), as listed in Table 2. Corresponding melt rates for the *av_melt* experiment are between 12.6 m yr$^{-1}$

and 54.3 m yr$^{-1}$. In both scenarios, pervasive thinning of all ice shelves causes significant inland migration of the grounding lines, as shown in Figure 4 for the *hi_melt* experiment, and Figure S1 for the *av_melt* experiment. The strongest retreat is concentrated along the central trunk of Pine Island Glacier, the Eastern Thwaites region, and Pope, Smith and Kohler East glaciers, which flow into the Crosson Ice Shelf.

| | Pine Island | | Thwaites | | Crosson | | Dotson | |
|---|---|---|---|---|---|---|---|---|
| | *hi_melt* | *av_melt* | *hi_melt* | *av_melt* | *hi_melt* | *av_melt* | *hi_melt* | *av_melt* |
| $m(t=0)\ \left[\text{m yr}^{-1}\right]$ | 64.4 | 54.3 | 46.4 | 40.9 | 22.4 | 12.6 | 27.3 | 21.3 |
| $T_{\star\text{IF}}(t=0)\ [^\circ\text{C}]$ | 2.5 | 2.02 | 2.35 | 2.07 | 1.94 | 1.62 | 1.96 | 1.46 |
| $\mu(t=0)$ | 1 | 1.15 | 1.22 | 1.26 | 1.18 | 1.03 | 0.95 | 1.06 |
| $\epsilon_T(t=0)$ | 0.7 | 0.7 | 0.72 | 0.72 | 0.63 | 0.61 | 0.73 | 0.72 |
| $\epsilon_U(t=0)\ \left[\text{m s}^{-1\circ}\text{C}^{-1}\right]$ | 0.002 | 0.0021 | 0.0011 | 0.0012 | 0.001 | 0.0011 | 0.0015 | 0.0018 |

**Table 2.** Average melt rates ($m$), thermal driving of the inflow across the ice front ($T_{\star\text{IF}}$), and transfer coefficients ($\mu$, $\epsilon_T$, $\epsilon_U$) at time 0 for the deep interior cavities, defined as areas of the ice shelf with draft below -400 m (Fig. 3). Note that in agreement with Eq. 10, $m \approx m_0 \epsilon_T \epsilon_U \mu^2 \left( T_{\star\text{IF}} \right)^2$.

Concomitant with changes in the cavity geometry, average melt rates in the deep interior respond to geometrical changes in

diverse ways, as can be seen from the top row in Figure 5. For the Crosson and Pine Island ice shelves, average melt rates in the





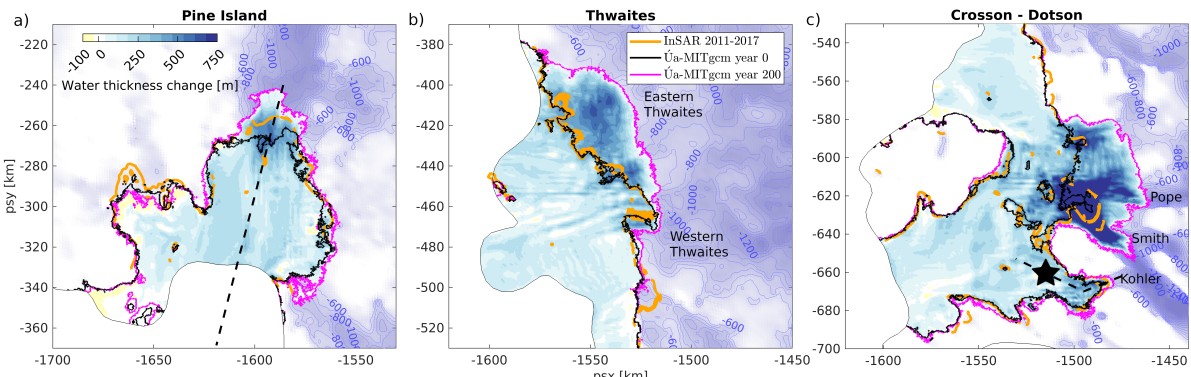

**Figure 4.** Change in water column thickness between years 0 and 200 of the *hi_melt* experiment. The Úa-MITgcm grounding line for both years is shown in black and magenta, respectively. For comparison, InSAR-derived grounding line locations between 2011 and 2017 (Rignot et al., 2016; Milillo et al., 2019, 2022) are shown in orange. Background colours and contours represent the bed topography (Morlighem et al., 2020). Dashed black lines in panels a and c correspond to the location of the thermohaline sections in Figure S2. The black star in panel c indicates the location of a bathymetric sill, discussed in Sect. 4.2

deep interior increase by as much as 75% and 35%, respectively. Average melt rates for the Dotson Ice Shelf decrease by up to 75%, whereas little to no changes occur for the Thwaites Ice Shelf. Whilst results for the *hi_melt* and *av_melt* simulations look broadly similar, the response of the latter is delayed due to the colder ocean thermal forcing and slower evolution of the cavity geometries. In the next 4 sections, the physical mechanisms that cause the varied response of basal melt to changes in

ice-shelf geometry will be explored. Specifically, the relationships between changes in cavity shape and different factors that determine the basal melt rate (thermal forcing at the ice front $\widetilde{T}_{\star\mathrm{IF}}$, the outer cavity transfer coefficient $\widetilde{\mu}$, the thermal transfer coefficient $\widetilde{\epsilon}_T$ and the momentum transfer coefficient $\widetilde{\epsilon}_U$ in Eq. 13) will be analysed in turn.

## 4.1 Thermal forcing at the ice front

Due to the two-way coupling between ice-flux divergence and basal melt, a rich pattern of ice-shelf thickness changes develops,

which evolves at a range of spatial and temporal scales. On average, the outer cavities in the *hi_melt* experiment expand vertically by 10 to 15m per decade, or between 200 and 300m over the 200-year duration of the simulation, as shown in Figure 4. This is consistent with present-day rates of ice-shelf thinning (Paolo et al., 2015; Adusumilli et al., 2018). The evolving water column thickness and variable freshwater input at the ice-ocean interface are coupled to changes in ocean currents and stratification in the cavities. These feedbacks can lead to significant variability in thermal driving of the inflow

at the ice front, despite the time-invariant restoring of ocean properties at the open boundaries of the domain, and despite the absence of open-ocean surface fluxes. At the start of the *hi_melt* simulation, the average thermal forcing of the deep inflow across the ice front ($T_{\star\mathrm{IF}}(t=0)$) ranges from 2.5 °C and 2.35 °C for the Pine Island and Thwaites ice shelves, to 1.94 °C and 1.96 °C for the Crosson and Dotson ice shelves (see Table 2). Values for the *av_melt* experiment are typically 0.3-0.5 °C lower.





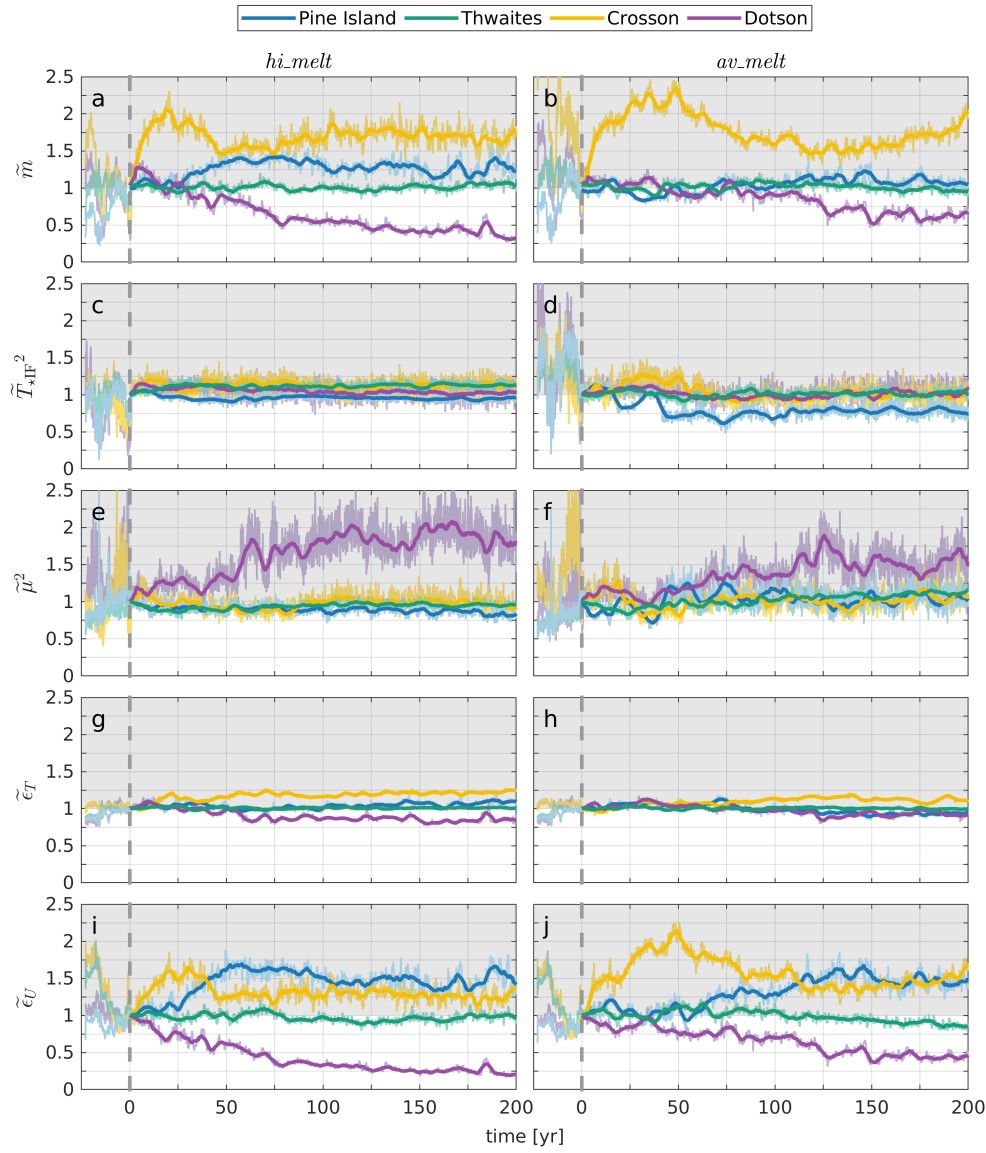

**Figure 5.** (a-b) Time series of average basal melt rates in the deep interior cavities for the *hi_melt* and *av_melt* experiments, normalized by values at time = 0. (c-d) Square of the normalized thermal forcing at the ice front, as defined in Eq. 12. (e-j) Time series of the normalized transfer coefficients, $\widetilde{\mu}$, $\widetilde{\epsilon}_T$ and $\widetilde{\epsilon}_U$, which link the thermal forcing at the ice front to melt rates in the deep interior according to Eq. 13. In all panels, thin lines correspond to monthly average values whilst overlaying thick lines correspond to a 5-year moving average. The dashed vertical line at time = 0 indicates the beginning of the *hi_melt* and *av_melt* experiments. Values at negative times correspond to an 18-year hindcast simulation with 1997-2014 ocean forcing, as described in App. A.

To diagnose changes in the thermal forcing of the cavity inflow and their impact on melt rates according to Eq. 13, a time series of $(\widetilde{T}_{\star\mathrm{IF}})^2$ is depicted in the second row of Figure 5. All cavities experience notable variability up to 25% at





a range of timescales. High-frequency fluctuations at monthly timescales are predominantly caused by Eddy activity at the ice front, which occur irrespective of the changes in cavity geometry, as is apparent from identical simulations but with a fixed cavity geometry (not shown). A detailed analysis of variability at sub(monthly) frequencies, including tidal variations, requires a higher sampling rate of the ocean data to avoid aliasing, and will not be pursued here. At longer, decadal to century

timescales, trends in $(\widetilde{T}_{\star\mathrm{IF}})^2$ are small, except for a 20% and 10% increase for the Thwaites and Crosson ice shelves in the *hi_melt* experiment, and a 25%-30% reduction for Pine Island Ice Shelf in the *av_melt* experiment. The long-term trends are a consequence of complex changes in cavity geometry, and difficult to attribute to individual processes. For example, the decrease in thermal forcing of Pine Island Ice Shelf coincides with a westward shift of the inflow pathways across the ice front and vertical contraction of the mCDW core, in conjunction with a reduced cyclonic re-circulation in the upper water column

of the outer cavity, which adjusts to migrating channels in the underside of the ice shelf.

Overall, the connection between variability in $T_{\star\mathrm{IF}}$ and melt rates in the deep interior in the *hi_melt* and *av_melt* experiments is weak. Indeed, if the time-varying factor $(\widetilde{T}_{\star\mathrm{IF}})^2$ in Eq. 13 is replaced by its (constant) value at $t = 0$, the Root Mean Square Error (rmse) between the resulting time series and $\widetilde{m}$ falls between 0.06 and 0.24 (see Fig. 6), which translates into relative errors in average melt less than 15%. The only exception is Pine Island Glacier in the *av_melt* experiment with rmse = 0.34,

which equates to a geometry-driven suppression of basal melt by 20-25%.

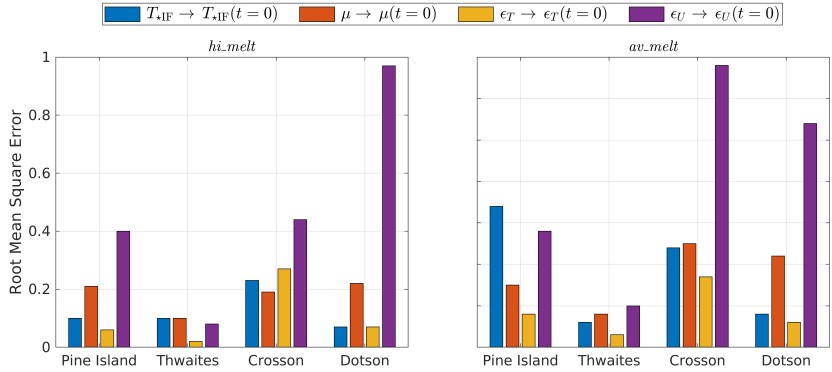

**Figure 6.** Root Means Square Errors (rmse) between $\widetilde{m}$ and the corresponding time series where a factor is replaced by its constant value at t=0: $\mathrm{rmse}\left(\widetilde{m}, \widetilde{m}\left(\mathcal{F} \rightarrow \mathcal{F}(t = 0)\right)\right)$, where $\mathcal{F}$ corresponds to $T_{\star\mathrm{IF}}$, $\mu$, $\epsilon_T$ or $\epsilon_U$. Rmse values provide an indication for how much of the temporal variability in $\widetilde{m}$ can be attributed to each factor. Further details are provided in Sect. 4 of the main text.

## 4.2 Outer cavity transfer coefficient

Thermohaline properties of the cavity inflow do not invariably propagate into the deep interior. An aggregated measure for how outer cavity processes can dampen or amplify changes in $T_{\star\mathrm{IF}}$ is provided by the outer cavity transfer coefficient $\mu$, defined in Eq. 11. At time $t = 0$, values of $\mu$ are between 1 and 1.18 (Table 2), which implies that the thermal forcing in the deep interior, 

$T_{\star\mathrm{DI}}$, is equal to or somewhat higher than $T_{\star\mathrm{IF}}$. With time, however, dynamics of the coupled ice-ocean system causes $\mu$ to





evolve in non-trivial ways. A time series of $\widetilde{\mu}^2$, which is the quantity of relevance for the calculation of the normalized melt rates in Eq. 13, is depicted in the third row of Figure 5. Values higher (lower) than one correspond to an increased (decreased) connectivity between the ice front and deep interior compared to the start of the simulations. Two key observations can be made.

Firstly, in both forcing scenarios the Dotson Ice Shelf experiences a strong amplification of thermal driving in the deep interior as the cavity geometry evolves. Although the grounding line does not migrate by more than 2 km (Figs. 4 and S1) and shoals due to its retreat up a steep prograde bed slope (see Figure S2 for a vertical section along the black line in Figure 4c), the ice draft rises by up to 300 m in places, which leads to a two-fold increase in water column thickness. As a result, the gap between the ice base and a bathymetric sill at the location indicated by the star in Figure 4c, is increased, which enables more

efficient transport of mCDW from the outer cavity into the deep interior (Fig. S2). This process, whereby shoaling of the ice draft enables warm waters to reach the grounding line, is similar to the mechanism that has been suggested to control the flow of mCDW towards the grounding line of Pine Island Glacier as it retreated from a bathymetric sill in the 1940s (De Rydt et al., 2014; Smith et al., 2017).

    The second observation concerns the existence of high-frequency, monthly variability in $\widetilde{\mu}$, which is particularly evident

for the Dotson Ice Shelf. For all ice shelves, these short-term fluctuations are strongly anti-correlated with monthly changes in $\widetilde{T}_{\star\text{IF}}$. The transfer coefficient $\mu$ therefore acts as a dampening factor for high frequency variability in the cavity inflow. Consequently, melt rates in the deep interior, as shown in Figure 5, are found to be relatively insensitive to short-term, Eddy-driven fluctuations of the thermal forcing at the ice front.

    The overall impact of changes in the outer cavity transfer coefficient on basal melt rates in the deep interior can be assessed

by replacing the time-varying factor ($\widetilde{\mu}^2$ in Eq. 13) by its value at $t = 0$. The rmse between the resulting timeseries and $\widetilde{m}$ are provided in Fig. 6, and fall between 0.08 and 0.25, which is equivalent to relative errors less than 10%. The only exception is the Dotson Ice Shelf, where the increase in thermal driving of the deep inflow causes a 50-60% increase in average basal melt when compared to conditions at the start of the simulations. However, it will be shown in Sect. 4.4 that despite the increase in ambient ocean thermal driving, melt rates of the Dotson Ice Shelf still decrease overall due to changes in the mixed layer

speed.

### 4.3 Thermal transfer coefficient

The third coefficient to impact on average basal melt rates in the deep interior cavities is $\epsilon_T$, defined in Eq. 6 as the ratio between the thermal driving of the deep inflow ($T_{\star\text{DI}}$ in Figure 3) and the thermal driving of the mixed layer adjacent to the ice-ocean interface ($T_\star$ in Figure 3). Values of $\epsilon_T$ at $t = 0$ years vary between 0.7 and 0.73 for Pine Island, Thwaites and Dotson

ice shelves (Table 2), and are somewhat lower for the Crosson Ice Shelf (0.61-0.63). Due to the input of cold freshwater at the ice-ocean interface, the thermal forcing of the mixed layer is strictly lower than that of the ambient ocean, or $\epsilon_T < 1$. It should also be noted that, for reasons that will be explored below, the value of $\epsilon_T$ is approximately independent of the forcing scenario.





In both simulations, the temporal variability of $\epsilon_T$ is less than $\pm 25\%$ (Fig. 5), with the largest changes found for the Crosson

and Dotson ice shelves. To understand the physical mechanisms that cause the increase or reduction of $\epsilon_T$ over time, it is instructive to analyze the relative importance, and underlying reasons, for variations in $T_{\star\mathrm{DI}}$ and $T_\star$. In Figure S3, a time series of $\epsilon_T$ for the *hi_melt* experiment is compared to $T_\star/T_{\star\mathrm{DI}}(t=0)$ and $T_\star(t=0)/T_{\star\mathrm{DI}}$. It follows that for all ice shelves, except the Dotson Ice Shelf, variations in $\epsilon_T$ are dominated by changes in the thermal forcing of the mixed layer, $T_\star$. This can be understood as follows. For a given thermal forcing of the inflow, variations in $T_\star$ are strongly linked to changes in the gradient

of the ice-shelf base, which controls the efficiency of turbulent entrainment of heat across the thermocline between the ambient water and mixed layer (Pedersen (1980); Jenkins (1991)). Indeed, as shown in Figure S3, changes in $T_\star/T_{\star\mathrm{DI}}(t=0)$ broadly track changes in the average gradient of the ice draft in the deep interior ($\nabla b$). The average basal gradient of the interior Crosson Ice Shelf cavity, for example, steepens from 0.034 to 0.054 during the 200-year *hi_melt* simulation. This coincides with a 28% increase in $T_\star$, whilst $T_{\star\mathrm{DI}}$ remains relatively constant and approximately equal to the thermal forcing at the ice

front ($\widetilde{\mu}^2 \approx 1$ in Figure 5e). As a result, $\epsilon_T$ increases from about 0.62 to 0.78.

Whilst values of $\epsilon_T$ are generally linked to the slope of the ice draft and hence the cavity geometry, for the Dotson Ice Shelf changes in $\epsilon_T$ are dominated by the previously highlighted increase in thermal driving of the inflow ($\widetilde{\mu}^2 \gg 1$ in Figure 5e). During the first 50 years of the simulation, this coincides with a rise in ice draft and flattening of the ice-base gradients, as can be seen from the thermohaline section in Figure S2. In turn, this causes a concurrent reduction in thermal driving of the mixed

layer (Fig. S3). Both increased thermal driving of the inflow and reduced thermal driving of the mixed layer cause the 20% reduction in $\epsilon_T$, observed in Figure 5g.

Whilst the foregoing discussion was primarily focused on results from the *hi_melt* experiment, results from the *av_melt* experiment were found to be qualitatively similar, but delayed in time, as suggested by Figure 5h.

Despite a demonstrable link between changes in ice-shelf draft and $\epsilon_T$, the relative impact of temporal variations in $\epsilon_T$ on

basal melt rates in the deep interior is small. When the time-varying factor $\widetilde{\epsilon}_T$ in Eq. 13 is replaced by its (constant) value at $t=0$, the rmse between the resulting timeseries and $\widetilde{m}$ falls between 0.02 and 0.17 (see Fig. 6), or a relative error in average basal melt rates less than 10%. The only exception is the Crosson Ice Shelf, with a rmse = 0.27, or a relative error around 20%.

## 4.4 Momentum transfer coefficient

The fourth and final coefficient to control average melt rates in the deep interior is the momentum transfer coefficient $\epsilon_U$, which

links the mixed layer velocity to the thermal driving of the inflow (Eq. 7). At decadal timescales, changes in $\epsilon_U$ dictate the majority of variability in the melt rates, as is apparent from the strong similarities between $\widetilde{\epsilon}_U$ and $\widetilde{m}$ in Figure 5. Since the temporal variability in thermal driving of the deep inflow is relatively small – the Dotson Ice Shelf being the only exception – changes in $\epsilon_U$ are dominated by variability in the mixed layer velocity $U_\star$. This is demonstrated in Figure S4, which shows a strong agreement between $\epsilon_U$ and $U_\star/T_{\star\mathrm{DI}}(t=0)$, whereas the similarity to $U_\star(t=0)/T_{\star\mathrm{DI}}$ is much weaker.

Based on a general scaling analysis, Holland et al. (2008) argued that for a given cavity geometry, $U_\star$ depends approximately linearly on the far-field thermal forcing. Results from the *hi_melt* and *av_melt* experiments suggest that the linear scaling factor, $\epsilon_U$, is strongly dependent on the geometry of the cavity. Whilst no studies have considered this dependency in detail, insights





from 2D plume theory ( Lazeroms et al. (2019)) and numerical ocean simulations for simplified ice-shelf shapes (Holland et al. (2008)) suggest that $\epsilon_U$ is a monotonically increasing function of the gradient of the ice base. However, analysis of the *hi_melt*

and *av_melt* experiments does not support such a monotonically increasing relationship. Instead, the two quantities are found to be weakly anti-correlated (not shown), indicating that $\epsilon_U$ cannot be quantified in terms of the geometry of the ice base alone, and furthermore, spatially averaged ice-shelf basal gradients only play a subsidiary role in controlling the mixed layer velocity, $U_\star$, at least for the complex geometries presented here.

Whilst basal gradients do locally impact on the buoyancy of the mixed layer and influence horizontal pressure gradients that

determine the geostrophic flow (Jenkins, 2016), the depth-average circulation is additionally constrained by the conservation of barotropic potential vorticity, henceforth referred to as BPV (Patmore et al. (2019) or Bradley et al. (2022)). To leading order, the depth-averaged flow aligns with contours of constant $\frac{f+\zeta}{H}$, where $f$ is the Coriolis parameter, which is approximately constant for the region of interest, $\zeta$ is the depth-averaged relative vorticity, and $H$ is the spatially variable water column thickness. As a result, the depth-averaged currents are strongly constrained by the (temporally evolving) 3D geometry of the

cavity.

An example of how the BPV and depth-averaged cavity circulation of Pine Island Ice Shelf co-evolve in the *hi_melt* experiment is provided in Figure S5. With time, BPV generally becomes less negative due to the increase in water column thickness $H$. At all times, no continuous contours of BPV exist that link the outer cavity to the deep interior, in part due to the existence of a prominent bathymetric sill that extends north-to-south (left-to-right in the figure) around $x = 300$ km (Jenkins et al., 2010).

As a result, there are no direct pathways of barotropic flow between the outer cavity and deep interior that conserve BPV, and the cavity circulation is divided into two cyclonic gyres (one in the deep interior, one in the outer cavity) separated by an anti-cyclonic current on the seaward side of the sill. The same current structure was previously reported by Dutrieux et al. (2014) and Bradley et al. (2022). As the grounding line retreats between years 0 and 60, the area with continuous contours of BPV landward of the sill doubles in size (the region indicated by the box with dashed lines in Figure S5), which allows the cyclonic

gyre to laterally expand. At the same time, the strength of the gyre increases from about 0.25 Sv to 0.65 Sv, sustained by the almost 2-fold increase in meltwater production (and associated increase in buoyancy) in the cavity interior, from 50 Gt yr$^{-1}$ to 90 Gt yr$^{-1}$. As the Pine Island grounding line retreats, the average gradient of the ice base does not significantly change (see section in Figure S2 along the dashed line in Figure S3a). Instead, the increase in meltwater production is enabled by, and sustained by, changes in the depth-averaged cavity circulation.

A more general analysis of the *hi_melt* and *av_melt* experiments shows significant temporal variability of the depth-mean flow in the deep interior of all cavities, as demonstrated in Figure 7 for the *hi_melt* experiment. Furthermore, a strong linear relationship exists between the depth-mean flow and the mixed layer speed, $U_\star$, and hence $\epsilon_U$. For the Pine Island Ice Shelf, a 75% increase of $\epsilon_U$ during the first 60 years of the simulation (Fig. 7a) followed by relatively constant values of $\epsilon_U$, coincides with a 2-fold increase and levelling off of the depth-mean flow, consistent with the discussion above. Both time series are

positively correlated with correlation coefficient 0.82. For the Thwaites Ice Shelf, neither $\epsilon_U$ nor the average depth-mean flow display any significant variability over the duration of the simulations. This does not contradict results from Holland et al. (2023), as the simulations presented here do not include the same detailed retreat of the Western Thwaites Ice Tongue. Instead,



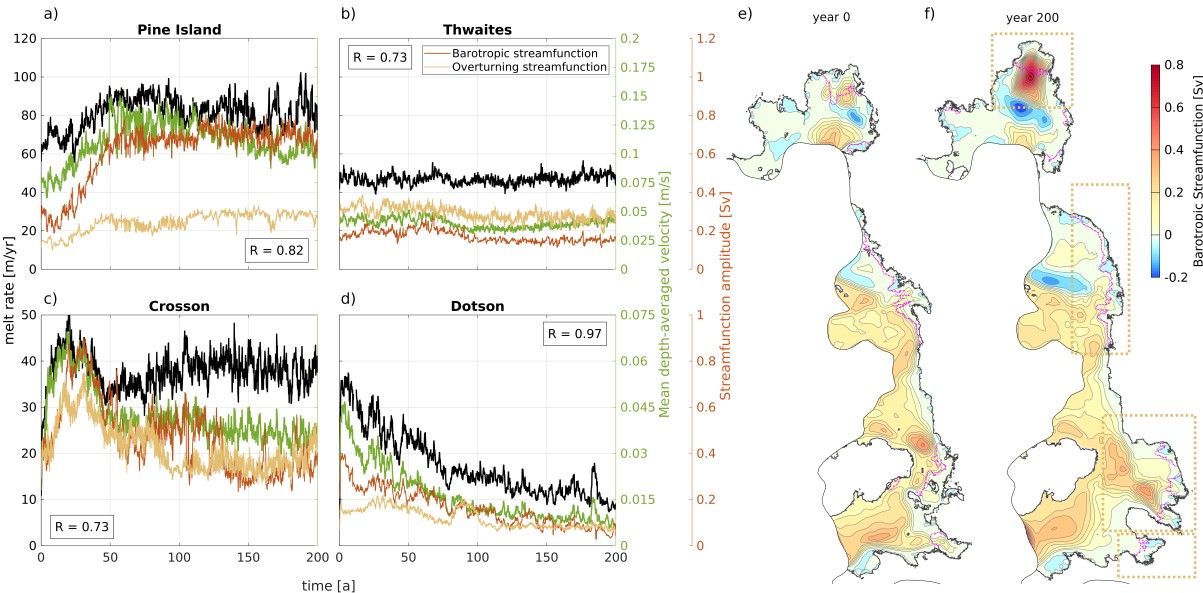

**Figure 7.** (a-d) Average melt rates (black lines, left y-axis) and average ocean current speed (green lines, right y-axis) in the deep interior cavities are shown alongside the barotropic and overturning streamfunction amplitudes (orange and yellow lines, right y-axis). The latter is calculated from the meridional (Pine Island) and zonal (Thwaites, Crosson, Dotson) transport within the boxes outlined by yellow dashed lines in panel f. $R$-values correspond to the correlation coefficient between mean melt rates and average current speed. (e-f) Barotropic streamfunction at years 0 and 200. Magenta lines delineate the boundaries of the deep interior cavities, i.e. areas with ice-shelf draft below -400 m. All plots are based on the *hi_melt* experiment.

at the start of the simulations, the Thwaites grounding line has already retreated past the pinning point that was highlighted by Holland et al. (2023) as the main geometric control on basal melt rates of the Thwaites Ice Tongue. The lack of any strong

geometrical feedbacks can be explained by the shallow ($> -900$ m) and relatively featureless bedrock topography over which the Eastern Thwaites grounding line retreats (Fig. 4). A narrow band of relatively low melt rates ($< 50$ myr$^{-1}$) tracks the moving grounding line (not shown) without significant modification to the geometrical configuration and currents. For the Crosson Ice Shelf, a sharp increase and decline of $\epsilon_U$ during the first 50 years is followed by approximately constant values for the remainder of the simulation. This coincides with equivalent changes in the depth-average flow (correlation coefficient 0.73).

The sharp increase in flow speed, and subsequent decline, are linked to the ungrounding of several large ice rises downstream of the main grounding line of Smith Glacier. The removal of these topographic barriers causes a reorganisation of the cavity circulation and melt rate distribution, which stabilizes after about 50 years, and the Smith Glacier continues its retreat into a deep, narrow trough. Finally, the gradual 4-fold decrease in $\epsilon_U$ for the Dotson Ice Shelf coincides with a 4-fold reduction in the depth-average flow (correlation coefficient 0.97). The slow-down of the circulation is largely explained by the loss of

buoyancy, as the grounding line retreats up a prograde bed slope and the ice draft rises above the thermocline in most places (Fig. S2).





An alternative way to analyze changes in the large-scale cavity circulation is through the barotropic streamfunction. The latter is presented in Figure 7 at $t = 0$ and 200 years, and corroborates the strengthening of the cyclonic gyre in the deep interior cavity of Pine Island Glacier, the relatively unchanged strength of the depth-integrated transport beneath the Thwaites Ice Shelf, a modest increase in circulation in the deep interior of the Crosson Ice Shelf cavity, and significant weakening of the circulation in the Dotson Ice Shelf cavity. Furthermore, the barotropic streamfunction amplitude, defined as the difference between the maximum and minimum value of the streamfunction in the deep interior (delineated by the magenta lines in Figure 7e-f) closely tracks the variability in $\epsilon_U$, as shown in panels a-d in Figure 7.

Together with the variability in horizontal transport, changes in basal meltwater production modulate the buoyancy-driven overturning circulation of the Amundsen cavities, as noted previously by Jourdain et al. (2017). Similar conclusions apply here. For each cavity, a measure of the overturning strength is obtained from the meridional (Pine Island) or zonal (Thwaites, Crosson, Dotson) overturning streamfunction amplitude, calculated for the areas enclosed by the boxes with dashed lines in Figure 7e. Timeseries of the baroclinic streamfunction, as shown in Figure 7a-d, indeed show temporal variability comparable to that of the barotropic streamfunction amplitude, with correlation coefficients $R = 0.84$ (Pine Island), $R = 0.57$ (Thwaites), $R = 0.73$ (Crosson) and $R = 0.78$ (Dotson).

Finally, as can be seen from panel j in Figure 5, results for the *av_melt* experiment are qualitatively similar, and are not explored in detail here. The only key difference compared to the *hi_melt* experiment is a delay in the melt-rate response due to the lower ocean thermal forcing and slower evolution of the cavity geometries.

## 5  The importance of far-field ocean variability

So far, geometrically driven changes in basal melt have been discussed in the context of suppressed variability of the far-field ocean. In reality, interannual to multi-decadal shifts in ocean conditions, linked to internal climate variability and anthropogenic trends in the atmosphere, are understood to be important regulators of ice-shelf melt in the Amundsen Sea. Aided by repeat measurements of ocean properties on the continental shelf, significant progress has been made in understanding the links between natural climate variability, including tropical ENSO activity, and decadal variations in basal melt (Jenkins et al. (2016) and reference therein, Silvano et al. (2022)). On the other hand, the impact of past and future anthropogenic climate change on ice-shelf melt has only recently started to be addressed (Naughten et al., 2022; Jourdain et al., 2022a). In all cases though, the interplay between different modes of far-field ocean variability and geometrically-driven changes in melt rates has not been included in numerical simulations. In this section, a first step is taken towards addressing this shortcoming. In particular, the combined impact of naturally occurring (present-day) ocean variability in the Amundsen Embayement and geometrically-driven changes in melt rates is considered. Uncertain anthropogenic trends in regional ocean conditions are not included here, but are the subject of a forthcoming study.

To compare geometrically driven changes in melt to changes caused by interannual to decadal variations in far-field ocean conditions, two numerical experiments were conducted, *ref_melt* and *var_melt*, as described in Sect. 2.2. In the *ref_melt* experiment, the ocean state was restored at the eastern and northern open boundaries to monthly average conditions over a 18-year



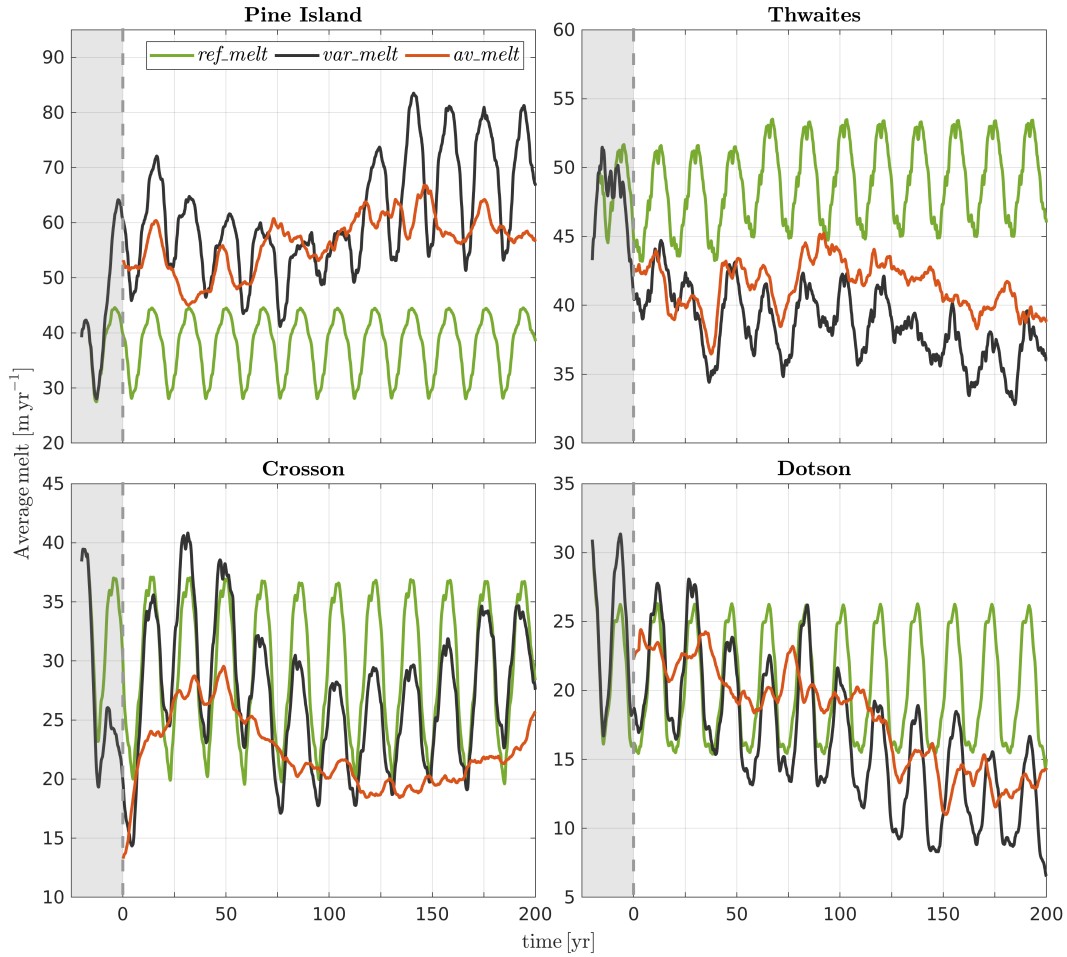

**Figure 8.** Five-year moving average of melt rates in the deep interior cavities for the *ref_melt* experiment (cyclic restoring of 1997-2014 ocean boundary conditions with static cavities), the *var_melt* experiment (cyclic restoring conditions with evolving cavities), and the *av_melt* experiment (constant restoring conditions with evolving cavities).

period (01/1997-12/2014), repeated 11 times, whilst the cavity geometry was kept fixed to its present-day configuration (see App. A for details about the geometry). A 5-year moving average of mean melt rates in the deep interior cavities, shown in Figure 8, displays 11 near-identical cycles with interannual variability ranging from 20% (Thwaites) to 50-60% (Pine Island and Dotson) and 80% (Crosson). The results for fixed ice-shelf cavities are plotted alongside results from the *var_melt* experiment, which imposes the same oceanographic restoring conditions but additionally includes evolving cavity geometries. Three

broad conclusions can be drawn.

1. The basal melt variability in the *var_melt* experiment consists of interannual changes superimposed onto a (multi)decadal trend. The latter is absent from the *ref_melt* experiment, but closely resembles the long-term variability in the *av_melt*





experiment (orange line in Figure 8), providing evidence for its geometrical origin. Results demonstrate that whilst basal melt variability is dominated by internal climate variability at decadal timescales, changes in cavity geometry can play

a non-negligible role at longer timescales. In particular, it should be noted that the impact of geometrical feedbacks can cause both a long-term amplification of basal melt, as is the case for the Pine Island Ice Shelf, and a long-term suppression of basal melt, as is the case for the Dotson Ice shelf.

2. Interannual changes caused by far-field ocean variability and multi-decadal trends caused by geometrical feedbacks are not independent. Indeed, the amplitude of interannual melt variability differs between individual 18-year cycles in

the *var_melt* experiment, which indicates that the sensitivity of melt rates to (inter)annual changes in far-field ocean conditions depends on the cavity geometry. This is particularly evident for the Pine Island Ice Shelf, showing lower amplitude variations in years 75-115, compared to, e.g., years 125-200. Whilst average melt rates scale approximately quadratically with the thermal driving of the inflow within each 18-year cycle (not shown), the scaling factor does not remain constant between cycles. This is consistent with the results presented in Sect. 4, which highlighted the strong

dependency of the scaling factor ($\mu^2 \epsilon_T \epsilon_U$ in Eq. 1) on the cavity geometry.

3. For the Pine Island and Thwaites ice shelves, the envelope of melt variability in the *var_melt* experiment falls outside the range of variability in the *ref_melt* experiment, underlining the dominant impact of geometrical feedbacks on the melt rates. Over the 200-year duration of the experiments, the average difference between the *var_melt* and *ref_melt* melt time series is 22 ma$^{-1}$ or 58% for the Pine Island Ice Shelf, -9.4 ma$^{-1}$ or -19% for the Thwaites Ice Shelf, -2.3 ma$^{-1}$ or -8%

for the Crosson Ice Shelf and -3.3 ma$^{-1}$ or -16% for the Dotson Ice Shelf. In all cases except for the Pine Island Ice Shelf, simulations that ignore geometrical feedbacks overestimate future melt rates forced by present-day ocean conditions. It remains to be shown that this conclusion holds in simulations with realistic atmosphere-ocean interactions and for weaker ocean restoring conditions that allow complex feedbacks between meltwater production, sea ice production and water mass properties on the continental shelf (Jourdain et al., 2017).

# 6   Glaciological implications

A prime objective of the preceding analysis was to identify potential feedbacks in the ice-ocean system, whereby grounding line retreat triggers changes in ocean dynamics with lasting implications for the ice-shelf mass balance, independent of the far-field ocean conditions. Whilst the focus has been on the response of ocean-driven melt to changes in cavity geometry, the impact of geometrical feedbacks on the net mass balance of the ice shelves, including grounding line fluxes, needs to be

considered in order to assess their importance for the future dynamics of the Antarctic Ice Sheet.

In broad agreement with the observational record (Depoorter et al., 2013), ice shelves in the Úa-MITgcm 1997-2014 hindcast simulation (Appendix A) have a negative net mass balance between -40 Gt yr$^{-1}$ and -10 Gt yr$^{-1}$, as shown by the black line for negative times (grey shaded area) in Figure 9. The interannual variability in mass changes during the hindcast period arises predominantly from changes in ice-shelf basal melting (solid an dashed blue lines in Fig. 9) due to variations in the



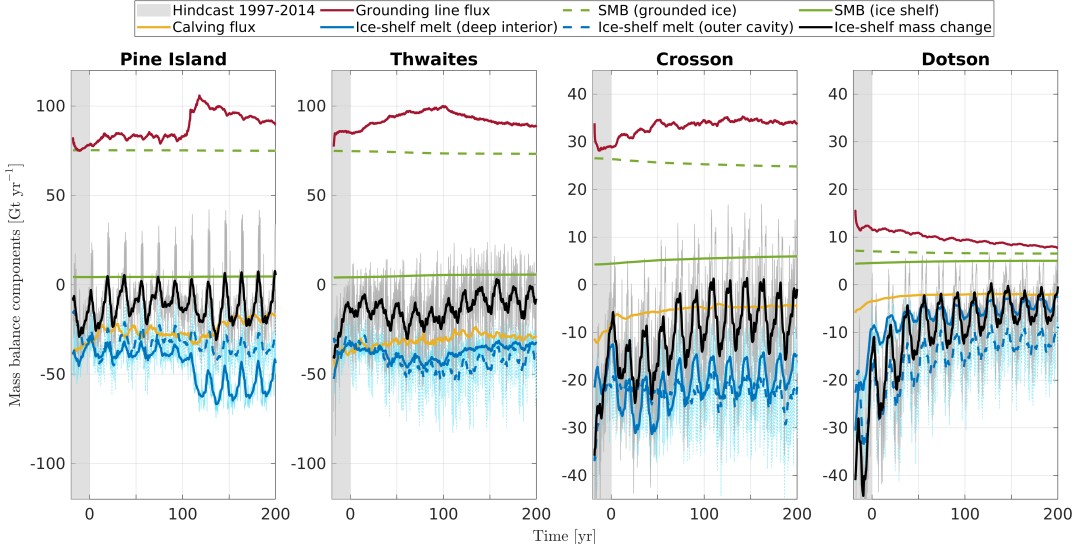

**Figure 9.** Temporal evolution of integrated mass balance components in the hindcast experiment (grey shaded area, see Appendix A for details) and the *var_melt* experiment (0 to 200 years, see Sect. 2.2 for details). Ice-shelf mass changes (black curve) are equal to the sum of the grounding line flux (red curve), surface mass balance (SMB, solid green curve), calving flux (orange line), ice-shelf melt in the deep interior (solid blue line), and ice-shelf melt in the outer cavity (dashed blue line). For reference the SMB of the grounded ice is represented by the dashed green line and is lower than the grounding line flux, indicating net mass loss (or positive sea-level contribution) from each glacier basin.

thermocline depth of the far-field ocean forcing. The corresponding variations in grounding line discharge (red line) and ice front flux (orange line) have a much lower amplitude.

It is prudent to recall that in all simulations, the surface mass balance (SMB, green lines) was kept constant and equal to a 1979-2015 climatology (van Wessem et al., 2018). While this assumption might not be valid in century-scale projections, present-day surface accumulation only contributes a small fraction of the total ice-shelf mass balance in the region, and any

impact of temporal variability on the net mass balance of the ice shelves will be disregarded. Furthermore, the ice front was fixed to its 1997 location, even though variations in ice-shelf extent and associated changes in buttressing have been shown to impact on the ice-shelf mass balance in recent years (Joughin et al., 2021; Bradley et al., 2022). Equally, changes in ice-shelf rheology, including rifting and fracturing, have been ignored, despite their importance for ice-shelf and ice-sheet dynamics in the region (De Rydt et al., 2021; Surawy-Stepney et al., 2023).

The focus in the remainder of this section will be on results from the *var_melt* experiment. Through repeat forcing of present-day decadal variability in ocean conditions and the inclusion of evolving cavity geometries, results from this simulation capture the combined impact of decadal ocean variability and geometrical feedbacks on the net mass balance of the ice shelves. Similar results for the *hi_melt* experiment with fixed ocean boundary conditions are provided in Figure S6, and will be discussed briefly at the end.



Results in Fig. 9 show that between years 0 and 200, the net mass balance of all ice shelves remains negative under the imposed ocean and atmospheric conditions. Only during intermittent episodes of 1 to 5 years does lower-than-average basal melting cause a positive net mass balance. Such episodes of positive net mass balance occur more frequently towards the end of the simulation, when ice-shelf mass changes tend to less negative values for the Thwaites, Crosson and Dotson ice shelves. Broadly speaking, fluctuations in basal mass loss, both in the deep interior and outer cavities (solid and dashed blue

lines in Fig. 9, respectively) dominate the variability in the net mass balance of the ice shelves at decadal timescales. The amplitude of variations in grounding line flux, surface mass balance and ice front (or calving) fluxes is significantly lower. At longer timescales, however, significant shifts in grounding line flux occur. One notable example is the rapid, 40% increase in grounding line discharge of Pine Island Glacier between years 100 and 120. For other ice shelves, changes in mass influx across the grounding line evolve more gradually, at least in the *var_melt* experiment, with a 25% (0.2 Gt yr$^{-2}$) increase over

100 years for the Thwaites Ice Shelf, a 15% (0.1 Gt yr$^{-2}$) increase over 50 years for the Crosson Ice Shelf, and a gradual 30% (0.02 Gt yr$^{-2}$) decline over 200 years for the Dotson Ice Shelf.

Without a compensating response of other components of the ice-shelf mass balance, the increase in grounding line flux for Pine Island, Thwaites and Crosson ice shelves would imply a smaller negative, or even positive net mass balance. For the Pine Island Ice Shelf in particular, the 40% (or 30 Gt yr$^{-1}$) increase in grounding line flux would lead to a persistently positive net

mass balance, causing the ice shelf to thicken and enable the glacier to readvance. Instead, the abrupt change in grounding line flux is largely compensated for by a shift in basal mass loss in the deep interior cavity, which enables the sustained thinning of the ice shelf and associated retreat of the grounding line. Similar findings apply to the other ice shelves: multidecadal trends in grounding line flux are largely compensated for by shifts in basal mass loss with similar magnitude but opposite sign. While changes in ice-front discharge occur too, they manifest themselves at longer timescales, and their general decline tends to shift

the net ice-shelf mass balance towards more positive values, rather than counteract the changes in grounding line flux.

The finding that melt rates play an important role in sustaining the negative mass balance of the ice shelves despite significant changes in grounding line flux, is not new. However, the physical mechanisms by which it does so, have not previously been studied in detail. In particular, mechanisms that enable sustained adjustments in basal mass loss that occur independent of the external forcing, have not been explored. In this study, the close interaction between changes in cavity geometry, ocean

circulation and basal melt has been identified as a candidate for such a mechanism. For example, based on the analysis in preceding sections, the 60% increase in meltwater flux in the deep interior of the Pine Island Ice Shelf cavity between 100 and 120 years, is enabled by a strong adjustment of the barotropic flow in response to changes in the cavity geometry. At the point where no further geometry-driven changes in circulation and melt rate occur, an increasing mass flux across the grounding line cannot be accommodated without causing a less negative or positive net ice-shelf mass balance. As a result, ice-shelf thinning

and grounding line retreat is either slowed down, or prevented. Results from the *var_melt* experiment indicate that this is more generally true: peaks in grounding line discharge coincide with a maximum in ice-shelf basal mass loss for all ice shelves, except the Dotson Ice Shelf. The evolution of the latter is somewhat different, as it experiences a gradual decline in grounding line flux, despite a persistent negative net mass balance of the ice shelf. Changes in grounding line flux are small, however, compared to the strongly negative trend in basal mass loss, which drives the net mass balance towards zero. The 75% reduction





in basal melt rates is again linked to changes in cavity geometry, as argued in Sect. 4, and drives the system towards a new equilibrium state with the grounding line of the tributary Kohler West Glacier (Fig. 4) located on a prominent bathymetric sill (Fig. S2).

Broadly similar conclusions can be drawn from the *hi_melt* experiment (Fig. S6) and the *av_melt* experiment (not shown). For both simulations, despite the absence of temporal variability in ocean forcing, sudden or gradual increases in grounding

line discharge into the Pine Island, Thwaites and Crosson ice shelves are largely compensated for by an increase in ice-shelf basal mass loss. Based on the analysis in preceding sections, such changes in basal melt can be linked to an adjustment of the cavity circulation in response to changes in the cavity geometry. Melt-geometry feedbacks therefore enable a sustained negative net mass balance of the ice-shelves, and promote ongoing retreat of the grounding line, independent of the far-field oceanographic conditions. As a final comment it is worth noting that whilst this mechanism operates under current glaciological

conditions, dynamical instabilities such as the marine ice sheet instability (Weertman, 1974, e.g.), can cause sustained retreat of the grounding line, independent of the amount of basal mass loss. However, no evidence for such a mechanism was detected here.

## 7 Summary and conclusions

Numerical simulations with a high resolution coupled ice-ocean model were used to quantify the connection and feedbacks

between changes in geometry and basal melting of ice shelves in the Amundsen Sea, West Antarctica. Under a range of present-day ocean conditions, sustained thinning of the Pine Island, Thwaites, Crosson and Dotson ice shelves was shown to cause pervasive retreat of the grounding lines by up to 50 km over a 200-year period. For each ice shelf, the resulting changes in geometry were linked to dynamical adjustments of the cavity ocean circulation, and to strong and varied changes in basal melt near the grounding lines. In particular, a significant correlation was found between the geometrical amplification (e.g.

Pine Island) or suppression (e.g. Dotson) of the barotropic cavity circulation, and the up-to 75% increase (Pine Island) or 75% decrease (Dotson) in average basal melt rate, irrespective of the far-field ocean forcing. The geometry-induced shifts in basal ablation were found to manifest themselves at decadal to multidecadal timescales, and outweigh, in some cases, variations in melt rates that occur due to natural variability in far-field ocean conditions. The newly-identified geometry-melt feedbacks play an important role in offsetting shifts in grounding line discharge, and thereby enable a sustained negative net mass balance of

the ice shelves, which drives further grounding line retreat and ice-sheet mass loss.

## Appendix A: Model initialization and validation

The Úa-MITgcm simulations presented in the main part of the paper were preceded by a spin-up and hindcast simulation, represented by the flow diagram in Figure A1. Given the abundant evidence that ice-sheet models, either in standalone mode (Seroussi et al., 2019) or coupled to an ocean model (Goldberg and Holland, 2022), retain information about their initial state

that can dominate the future evolution of the ice sheet, and the fact that diverse approaches to (coupled) model initialization are




used in the literature, a detailed account of the Úa, MITgcm and Úa-MITgcm initialization procedures are provided in Sect. A1. A basic validation of the coupled configuration, based on a 1997-2014 hindcast simulation, is provided in Sect. A2.

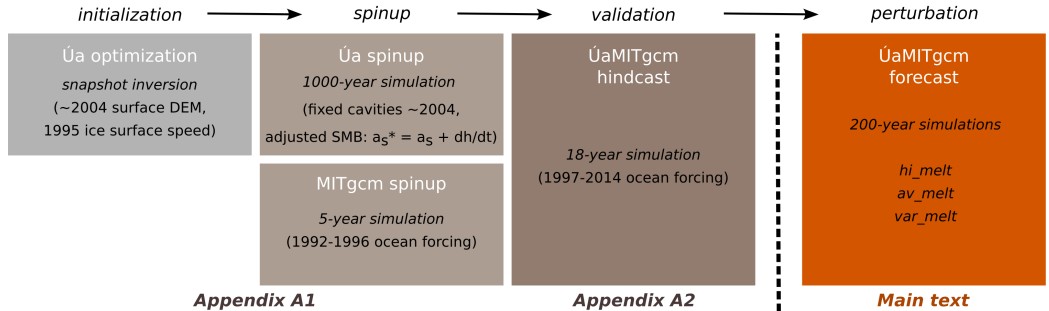

**Figure A1.** Flowchart showing the different steps in the intialization, spinup and validation of the coupled ice-ocean model Úa-MITgcm. Details about the initialization and spinup of the standalone Úa and MITgcm configurations are provided in Sect. A1. Results from a Úa-MITgcm hindcast simulation between 1997 and 2014 are provided in Sect. A1. Results from the 200-year perturbation experiments (*hi_melt*, *av_melt* and *var_melt*) are presented and discussed in the main part of the text.

## A1 Initialization and spinup

As a first step in the initialization of the coupled Úa-MITgcm configuration, the ice flow model Úa was calibrated using an inverse method, first introduced by Macayeal (1992) and now frequently used in ice-sheet modelling (e.g. De Rydt et al. (2021); Barnes et al. (2021)). The inverse step resulted in optimal estimates of the spatially varying rate factor ($A$) in Glen's law and the slipperiness ($C$) in the basal sliding law, for a given ice-sheet geometry and surface velocity measurements ($u_{\mathrm{obs}}$) with corresponding errors ($\varepsilon_u$). Values for the $A$ and $C$ fields were obtained as a solution to the minimization problem $\mathrm{d}_{[A,C]} J = 0$, where the cost function $J$ was defined as the sum of the misfit terms for the surface velocities and ice thickness changes, and a Tikhonov regularization term: $J = I + R$, with

$$I = \frac{1}{2\mathcal{A}} \int \mathrm{d}\mathbf{x} \left( (u_{\mathrm{model}} - u_{\mathrm{obs}})^2 / \varepsilon_u^2 + (\partial_t h|_{\mathrm{model}} - \partial_t h|_{\mathrm{obs}})^2 / \varepsilon_{\partial_t h}^2 \right) \tag{A1}$$

$$R = \frac{1}{2\mathcal{A}} \int \mathrm{d}\mathbf{x} \sum_i \left( \gamma_{i,s}^2 \left( \nabla \log_{10} (p_i/\hat{p}_i) \right)^2 + \gamma_{i,a}^2 \left( \log_{10} (p_i/\hat{p}_i) \right)^2 \right), \tag{A2}$$

where $\mathcal{A} = \int \mathrm{d}\mathbf{x} I$ is the total area of the domain, $p_i$ and $\hat{p}_i$ with $i \in [1,2]$ are shorthand notation for $p_1 = A$, $p_2 = C$ and their respective priors $\hat{p}_1 = 5 \times 10^{-9}$ yr$^{-1}$kPa$^{-3}$, $\hat{p}_2 = 0.0015$ m yr$^{-1}$kPa$^{-3}$. Values of the pre-multipliers $\gamma_{A,s} = \gamma_{C,s} = 10^5$ and $\gamma_{A,a} = 250$, $\gamma_{C,a} = 1$ were chosen based on a common L-curve approach (De Rydt et al., 2021). The velocity observations ($u_{\mathrm{obs}}$) and corresponding errors ($\varepsilon_u$) in the first misfit term were taken from the ERS-1/2 InSAR tandem mission in late 1995 - early 1996 (Rignot et al., 2004). Ice thickness changes were set to zero everywhere ($\partial_t h|_{\mathrm{obs}} = 0$) with errors $\varepsilon_{\partial_t h} = 0.1$ and $\varepsilon_{\partial_t h} = \infty$ for grounded and floating ice, respectively. The ice-sheet geometry was assembled from a combination of Bedmachine Antarctica v2 data for the bed geometry (Morlighem et al., 2020), and the Icesat-corrected surface DEM from



the ERS-1/2 mission with effective timestamp in 2004 (Bamber and Griggs, 2009). Nominal ice densities ($917 \ \mathrm{kg \ m^{-3}}$) were corrected using a climatology of the firn density and thickness distribution from the RACMO2.3 dataset between 1979 and 2013 (van Wessem et al., 2018). The inversion for $A$ and $C$ was stopped after 1000 iterations with a mean and standard deviation of the misfit between modelled and observed surface speed of $12 \ \mathrm{m \ yr^{-1}}$ and $61 \ \mathrm{m \ yr^{-1}}$, respectively.

Following the inversion step, the transient evolution of the ice-sheet can be contaminated by unrealistic short-wavelength ($< 10$ km), large-amplitude ($> 10$ m/yr) adjustments that dominate changes in ice thickness during the first few years of the simulation. These adjustments arise from a force imbalance due to inconsistent and uncertain input data, the simplified physical description of ice flow, and errors in the external forcing such as the surface and basal mass balance. Importantly, the initial transient modes can stimulate the growth of unrealistic dynamic behaviour, which can drive the solution away from the observed trends. For example, a localized, short-lived increase in ice thickness at the grounding line can cause an advance of the ice sheet, and induce a positive feedback between slow-down and further advance, independent of the observed trends. To dampen the spurious changes in ice thickness, whilst preventing the geometry to drift away from its present-day configuration, a spinup of the ice sheet was carried out with Úa in standalone mode (Fig. A1), following a method similar to Arthern and Williams (2017). The ice velocity and thickness were evolved for 1500 years with two constraints: the ice thickness was fixed to its initial value for all nodes that were afloat at time 0, and the surface mass balance of the grounded ice was modified by an additional term, $as \to as - \partial_t h$, where $\partial_t h$ corresponds to the spatially variable, time-average rate of ice thickness change between 1997 and 2001 from satellite measurements (IMBIE Team, 2018). After 1500 years, the ice sheet reached an approximate steady state, with a mean change and standard deviation in grounded ice thickness and surface velocities of $-12.7 \pm 54$ m and $-13.5 \pm 55 \ \mathrm{m \ yr^{-1}}$ compared to the initial state. Importantly, the ice-shelf extent, and therefore the grounding line location, remained approximately unchanged during the spinup simulation, with only a small inland migration in isolated areas. Moreover, any simulation that is started from the spinup state with surface mass balance $as$, will reproduce initial changes in grounded ice thickness that correspond to the observed rates of change in the late 1990s ($\partial_t h$ above).

The ice-sheet geometry at the end of the Úa spinup was transferred to MITgcm, and used in a five-year, ocean-only, spinup with static ice shelf cavities. This allowed the ocean state to adjust from its uniformly stratified initial conditions to a dynamic state in equilibrium with the varying boundary conditions. The latter correspond to 1992-1996 monthly mean values of temperature, salinity and velocity from Kimura et al. (2017), linearly interpolated onto the open ocean boundaries of the regional MITgcm configuration (Fig. 1). Given typical flow speeds of $0.01 \ \mathrm{m \ s^{-1}}$, a five-year spin-up period was considered adequate for the 450 km by 250 km ocean domain with relatively small ice shelf cavities. At the end of the ocean spinup (31/12/1996), the coupling between Úa and MITgcm was switched on (Fig. A1), and the coupled configuration was stepped forward with monthly-mean ocean restoring conditions between 01/01/1997 and 31/12/2014, based on Kimura et al. (2017). Results from this hindcast experiment, referred to as *var_melt*, are presented in the next section.

## A2    Model validation

The performance of Úa-MITgcm in its configuration specific to this study, is assessed by comparing model output to available Earth Observation and in situ ocean data between January 1997 and December 2014. Whilst model validation is a crucial step



towards certifying the model's ability to accurately predict future changes in Antarctic ice volume, a comprehensive validation

exercise is not attempted here. It is recognised that further optimization of the current configuration is required in order to eliminate some of the biases that could play a distinctive role in future ice loss projections.

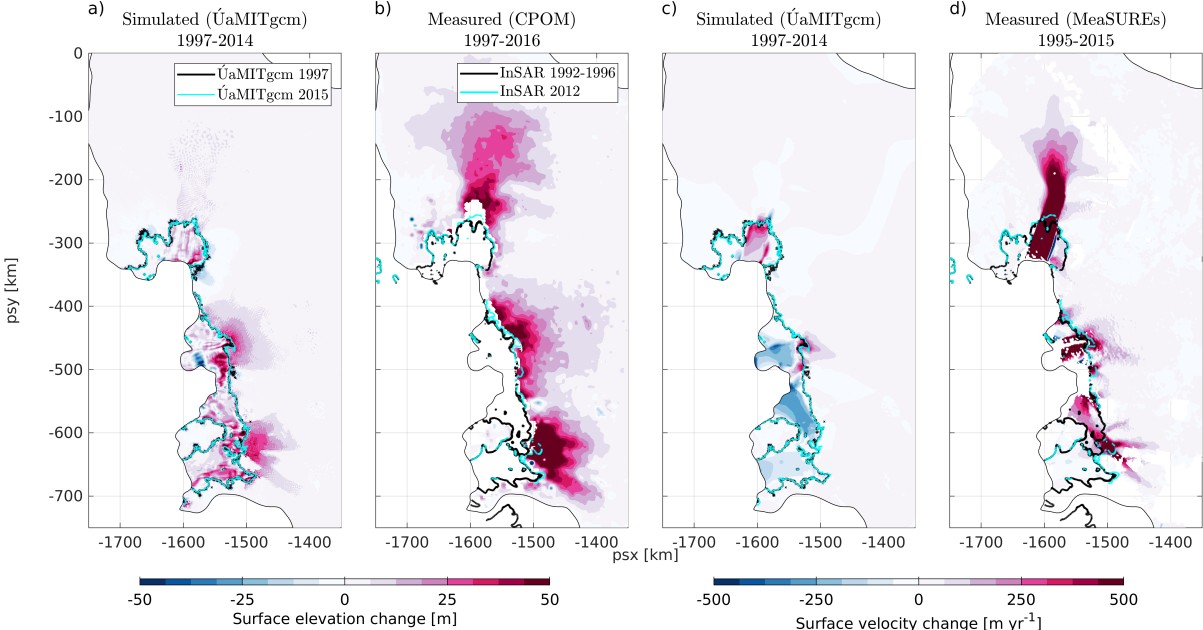

**Figure A2.** A comparison between simulated (a, c) and measured (b, d) changes in surface elevation change and surface velocity changes between 1997 and 2015. Simulated results are from the coupled Úa-MITgcm hindcast experiment *var_melt*, which starts from an initial ice-sheet geometry representative of the late 1990s - early 2000s, and is forced by ocean restoring conditions between 1997 and 2015 (Kimura et al., 2017). Observed changes in surface elevation of the grounded ice between 1997 and 2016 (panel b) were obtained from CPOM (IMBIE Team, 2018); changes in surface velocity (panel d) were provided by the MeaSUREs project (Rignot et al., 2017). Grounding line locations in 1997 and 2015 from Úa-MITgcm are shown in panels a and c in black and cyan, respectively. Corresponding estimates from InSAR data (panels b and d) were taken from Rignot et al. (2016).

A comparison between modelled and observed changes in ice-sheet surface elevation and surface velocities between 1997 and 2015 are presented in Fig. A2. Whilst Úa-MITgcm simulates widespread negative changes in ice thickness with a spatial pattern that matches the observations, its magnitude and accompanying upstream migration of the grounding line between

1997 and 2015 is significantly underestimated. In agreement with the slower-than-observed changes in ice thickness, changes in surface velocity are smaller than observed. For all ice shelves except Pine Island Ice Shelf, the simulated change in surface speed has the opposite sign compared to observations. Whilst the sign and magnitude of the observed changes is ultimately simulated at later times in the different Úa-MITgcm perturbation experiments (*hi_melt*, *av_melt* and *var_melt* in Sect. 2.2), the ice-sheet response is delayed and depends strongly on the ocean forcing. The slower-than-observed response can be caused by

a variety of poorly-constrained or missing processes, including but not limited to the form of the basal sliding law, the lack





of ice-shelf damage evolution and calving front migration, and a potential misrepresentation of ocean-induced basal melting in shallow cavities with unknown bathymetry near the grounding line. A quantification of the uncertainties related to these processes, and further optimization of the ice-ocean model will be pursued in future studies. In particular, the construction of an initial state that minimizes the misfit between simulated and observed trends in ice thickness and speed over the observational period is an essential requirement for the correct timing of any future mass loss from the West Antarctic Ice Sheet (Goldberg and Holland, 2022).

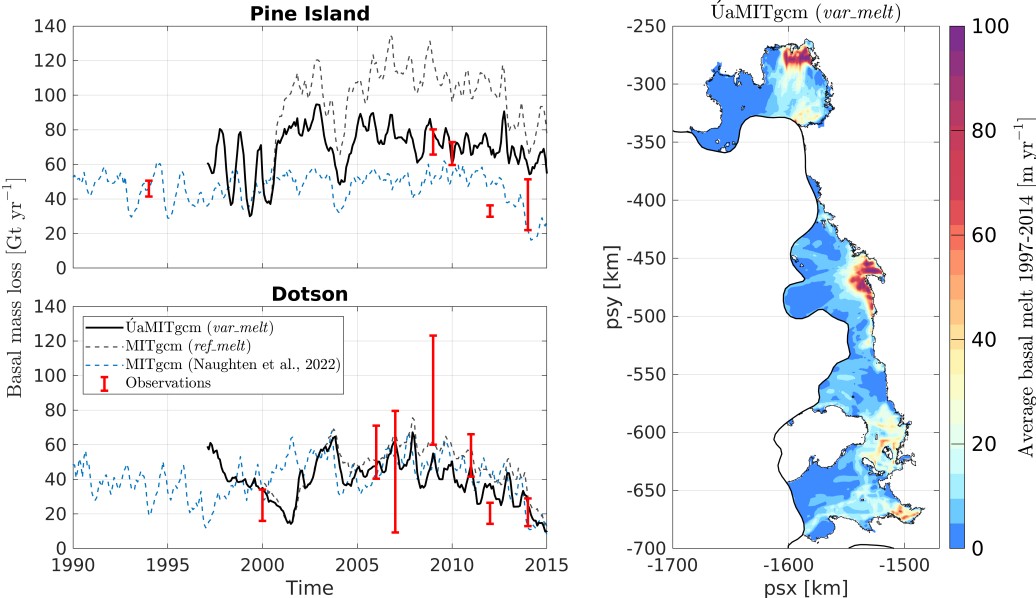

**Figure A3.** A comparison between simulated and measured net basal mass loss for the Pine Island and Dotson ice shelves. Results from the coupled Úa-MITgcm hindcast simulation (*var_melt*) and two stand-alone MITgcm simulations with fixed cavities (*ref_melt* and results from Naughten et al. (2022)) are compared to observations for the Pine Island Ice Shelf (years 1994, 2009, 2010 and 2012 from Dutrieux et al. (2014) and 2014 from Heywood et al. (2016)) and the Dotson Ice Shelf (Jenkins et al., 2018). The right panel shows average melt rates between 1997 and 2014 from the *var_melt* simulation.

A comparison between simulated and observed ice-shelf melt rates is provided in Fig. A3. Overall, integrated meltwater fluxes from the coupled hindcast experiment (*var_melt*) agree well with observations, except for Pine Island Ice Shelf in 2012 and 2014, when the model overestimates basal mass loss by about 50%. The temporal variability of *var_melt* compares favorably to results from Naughten et al. (2022) (dashed blue line in Fig. A3), which include a more complete treatment of ocean-atmosphere interactions. The agreement is not surprising given the current setup uses restoring conditions at the ocean boundaries that were obtained from a regional ocean simulation by Kimura et al. (2017), which has a similar model architecture as Naughten et al. (2022), and was forced by ERA-Interim atmospheric conditions that are similar to the improved ERA5 atmospheric conditions used in Naughten et al. (2022). The good agreement also holds true despite the absence of ocean



surface fluxes in this study, which is an indication that the latter do not strongly affect the basal melt variability. Any remaining discrepancies between the current Úa-MITgcm configuration and Naughten et al. (2022), in particular for Pine Island Glacier, can be caused by a number of differences in ocean parameter choices, cavity geometry, boundary forcing, model resolution etc., and a detailed comparison will not be pursued here. Results from a stand-alone MITgcm simulation with the same 1997-2014 ocean boundary restoring conditions but with fixed ice-shelf cavities (*ref_melt* experiment, dashed black line in Fig. A3),

produced melt rates for Pine Island Ice Shelf about 50% higher than results from the corresponding *var_melt* experiment with evolving cavities. This highlights the important impact of changes in cavity geometry on basal melt rates, and the realization that the calibration of model parameters against observations can lead to distinctly different parameter choices for stand-alone and coupled setups.

The spatial distribution of basal melt in the *var_melt* experiment, averaged between 1997 and 2014, is shown in Fig. A3.

Average melt rates in excess of 100 m yr$^{-1}$ are found near the grounding lines of Pine Island and Thwaites glaciers, while basal ablation of the Crosson and Dotson ice shelves is generally lower due to lower thermal driving of the far-field ocean (Sect. 4.1). While satellite observation-based estimates of basal melt rates are available for comparison ( Gourmelen et al. (2017) and Adusumilli et al. (2018)), results are not reproduced here. Significant data gaps remain in regions with the highest melt rates near the grounding lines of Pine Island and Western Thwaites ice shelves, making it difficult to validate the simulated melt

patterns in those regions.

*Author contributions.* JDR designed and initiated the project, performed the model simulations, carried out the analysis, and produced the figures and manuscript. KN coded the latest version of Úa-MITgcm and reviewed and edited the paper.

*Competing interests.* Jan De Rydt serves as topical editor for EGUSphere and The Cryosphere.

*Acknowledgements.* Jan De Rydt was supported by the TiPACCs project, which receives funding from the European Union's Horizon

2020 research and innovation programme under grant agreement no. 820575. Kaitlin Naughten is supported by the NERC LTSM project TerraFIRMA.



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
