# Peer review of "Geometric amplification and suppression of ice-shelf basal melt in West Antarctica"

_EGUsphere, 2023_

## Author Comment (AC1)

**Reply to reviewers**

We would like to thank both reviewers for their detailed and constructive feedback. In their general comments, they raised broadly similar points, which we collated and address together, followed by a point-by-point reply to their technical comments. We also included a new version of the manuscript with tracked changes at the end of this document. All original reviewer comments are in black; replies in ***bold italic blue*** and changes to the manuscript in *italic blue*.

**Main points**

*Reviewer 1 says:* "The paper leaves the reviewer with a mixed impression. The study design and methodology are state of the art, figures are well crafted, and the presentation is mostly clear. There is, however, still some room for improvement."

*Reviewer 2 says:* "The discussion and interpretation of results are thorough and with appropriate consideration of related work. Explanations are detailed and the manuscript maintains a logical flow. Despite the complexity, the overall presentation is mostly clear, enhanced by well-structured and colorblind-friendly figures. Certain minor aspects could benefit from refinement for enhanced clarity and depth. .... The entire analysis and the overview of the three numerical experiments are clear and concise. I have only 3 suggestions."

***We thank both reviewers for their overall positive assessment and welcome their suggestions for improvements. We are pleased to know they found the methodology to be sound, and the presentation to be logical and clear. Below is a point-by-point reply to their main points and our amendments to the manuscript.***

1. *Reviewer 1 says:* "First, the definition of cavity transfer coefficients gives the study a quantitative touch. That's nice, but while the coefficients are defined by the equations in which they appear, the paper is very short on information on how they are actually computed from the model results."

   *Reviewer 2 says:* "The authors focus on the different cavity transfer coefficients in order to diagnose the feedbacks between changes in basal melt, imposed ocean boundary conditions and cavity geometry. This deserves to be highlighted and it could be beneficial to provide more information on how these cavity transfer coefficients are computed."

   ***We have added a paragraph in section 3 to explain, in general terms, how the transfer coefficients were computed from MITgcm output:***

   *"To calculate the quantities in Eq. (13) or, equivalently, Eqs. (10) and (11), from model output requires a discrete version of the equations. In MITgcm, all variables in Eq. (10), including m, $U_*$ and $T_*$ are defined at cell centers of the Arakawa C grid, and spatial integrals directly translate into discrete sums. The calculation of $T_{*DI}$ and $T_{*IF}$ in Eq. (10) requires the evaluation of line integrals along the -400 m ice draft contours and ice fronts, respectively. Draft contours were calculated from the discrete MITgcm geometry, leading to a connected sequence of line segments with nodes that coincide with MITgcm nodes or the midpoint between MITgcm nodes. Ocean state variables and geometric variables were linearly interpolated to the midpoint of each line segment; velocities were subsequently projected to obtain the flow perpendicular to each segment. The contour integral was then replaced by a sum over all line segments, and quantities evaluated at the center of each segment. For the calculation of $T_{*IF}$ at*

*the ice front, a similar procedure was followed, but contour nodes were taken directly from the Ua mesh. Further details on how each transfer coefficient was calculated from raw MITgcm output can be found on GitHub. A link to the code has been provided in the data availability statement at the end of the paper."*

*We did not include the discretized version of each equation in the main text, since the technical details are specific to the numerical architecture of the model and are not directly transferrable to other studies. This allowed us to keep a tighter focus on the physical processes. Instead, we have referred the reader to our GitHub repository, which contains all the scripts that are required to calculate the coefficients from raw MITgcm output, as well as the code needed to produce the associated figures in the manuscript.*

*In addition, to better guide the reader through section 3, we have split the section into 6 subsections, with explicit reference to MITgcm in the title of subsection 3.6: 3.1. Basal melt parameterization, 3.2. Thermal transfer coefficient, 3.3. Momentum transfer coefficient, 3.4. Basal melt in the deep interior, 3.5. Outer cavity transfer coefficient, 3.6. Summary and application to MITgcm.*

2. *Reviewer 1 says:* "Furthermore, the coefficients are not used to their full potential, and actually in two ways so. Firstly, the authors provide time series of all coefficients for all ice shelves for many of the experiments, but we do not learn much about the processes that shape the feedback in different ways for different ice shelves. There is a lot of potentially interesting analysis hidden in the discussion of coefficient time series (e.g. on how sub-ice circulation responds to changes in geometry), but the discussion goes coefficient-by-coefficient where going ice shelf-by-ice shelf would have made it easier to actually learn things about the real world. The authors are therefore encouraged to re-organize this section."

*This is a valid point, and something we have pondered when drafting the manuscript. We did not fully reorganize section 4, as the reviewer suggested, but instead subdivided section 4.5 (formerly 4.4) into clearly marked paragraphs for each ice shelf, for the following 2 reasons:*

*1. The paper's main conclusion is that temporal changes in melt rates near the grounding line are dominated by changes in the momentum transfer coefficient. A much smaller proportion of the variability is explained by the heat transfer and outer cavity transfer coefficients. Importantly, this conclusion is independent of the ice shelf, and highlights a common, dominant physical process that does not depend on the specific geometrical setting. To underscore this point, we structured section 4 according to the different physical processes that affect the melt rates (i.e. the transfer coefficients), and analyzed common behavior, rather than treating each ice shelf as a separate entity.*

*2. We conclude that the geometrically driven changes in barotropic and overturning circulation, quantified through the momentum transfer coefficient, dominate ice-ocean feedbacks for all ice shelves. Whilst we find that, depending on the geometrical characteristics, this process can lead to suppression or amplification of the melt rates, the details of how this will pan out depends on the far-field ocean forcing. Since our simulations are forced by idealized far-field conditions, we cannot draw definite conclusions about the evolution of each ice shelf under future climate scenarios. We focus instead on identifying the dominant physical processes that control melt-geometry feedbacks (i.e. the cavity transfer coefficients) and have structured section 4 accordingly.*

*In order to better convey the rationale behind this choice, we made the following changes to the manuscript:*

- *We rewrote the paragraph preceding subsection 4.2 (formerly 4.1) as follows:* *"In the next 4 subsections, the physical processes that cause the varied response of basal melt to changes in ice-shelf geometry are explored. In particular, the evolution of the thermal driving at the ice front ($T_{IF}$) and 3 transfer coefficients ($\mu$, $\tilde{e}_T$, $\tilde{e}_U$) is analyzed one-by-one, allowing to draw general conclusions about the physical processes that control melt-geometry feedbacks, independent of the specific geometric characteristics of each ice shelf. As will be shown in section 4.5, the momentum transfer coefficient $\tilde{e}_U$ dominates the melt evolution, and a more in-depth analysis of the underlying dynamical causes is provided for each ice shelf."*

- *We changed the titles of subsections 4.2-4.5 (formerly 4.1-4.4) to better indicate the physical mechanism and role of each transfer coefficient:*

  *4.2 "Limited change in thermal forcing at the ice front"* **instead of "Thermal forcing at the ice front"**

  *4.3 "Geometrically constrained transport of water masses from the ice front to the interior cavity"* **instead of "Outer cavity transfer coefficient"**

  *4.4 "Thermal forcings of the ice-ocean boundary layer and deep inflow are linearly related"* **instead of "Thermal transfer coefficient"**

  *4.5 "Changes in cavity circulation dominate melt rate evolution"* **instead of "Momentum transfer coefficient"**

- *We rewrote section 4.5 (formerly 4.4) to include a more balanced discussion about the barotropic and overturning circulation (see our reply to later comments by reviewer 1) and to add clearly marked paragraphs for each ice shelf. In particular, an ice-shelf-by-ice shelf description of the dynamical evolution of the cavity circulation is provided, giving insights into the physical processes that dominate changes in melt for each cavity. A new version of section 4.5 can be found in the manuscript with tracked changes at the end of this document.*

3. *Reviewer 1 says:* "Secondly, the authors motivate their study by going „towards bridging the gap between [two] incomplete modelling approaches", namely fixed-geometry melt rate modelling on one side, and ice sheet modelling with weakly constrained (and not always physics-based) melt rates on the other side. Given that coupled ice sheet-ocean models do exist (one is in this paper) but with the currently available computational resources cannot even dream of fully covering the long timescales relevant to ice sheet processes, there is clear scope for guiding ice sheet modelling communities towards how to represent geometry-melt rate feedback with more physics than typically used in ice sheet models now, which creates a strong motivation for a study like the one presented – but unfortunately the authors do not follow up on this aspect. Calling for a full set of guidelines that modellers can follow would be asking too much, but a little bit of perspective on how exactly the results of this study can „be considered in future projections of Antarctic mass loss" [quote from the abstract] would make this paper a lot stronger."

*Reviewer 2 says:* "Overall, the summary & conclusions part could benefit from a slightly more detailed overview of the findings. It would be good to highlight the implications more and to mention specifically how their work overcomes the limitations of current ocean modeling approaches in the context of the Amundsen Sea's basal melt rates. The results and interpretations of the findings are well documented and highlighting them in the conclusions would provide a clearer and more impactful closure to the paper.

*We agree with both reviewers that the summary and conclusions could better highlight the main findings of the paper and make stronger statements about the implications of our results. We have rewritten this section to better reflect the key results and added words of caution against the use of parameterized melt rates in stand-alone ice sheet models. We were reluctant to call these 'guidelines', as reviewer 1 suggest, because any recommendation for best practice should be thoroughly tested in stand-alone ice-sheet models, which is not something we set out to achieve in this paper. The new summary and conclusions are included in the manuscript with tracked changes at the end of this document.*

4. *Reviewer 2 says:* When describing the model's domain and specifications (in "2.1. Coupled ice-ocean model setup"), consider adding a brief explanation of why these specific parameters and configurations were chosen. It could enhance the clarity for less experienced readers and it would provide context for the approach. For example, is the resolution and number of layers selected purely based on the compromise between computational feasibility and accuracy or was the nature of specific physical processes considered as well?

*We have added further motivation for our choice of model configuration in section 2.*

*For MITgcm:*

*"... The northern and western edges of the domain were chosen to fully cover the present-day extent of the aforementioned ice shelves and their respective mCDW inflow pathways but exclude the adjacent Cosgrove Ice Shelf in the east and Getz Ice Shelf in the west. MITgcm solves the Boussinesq and hydrostatic form of the Navier Stokes equations on an Arakawa C grid in polar stereographic coordinates with a uniform Eddy-permitting horizontal resolution of 1.3 km. The z-coordinate levels consist of 80 layers with 20 m resolution down to a depth of -1600 m, and 10 layers with 40 m resolution down to -2000 m. Although the seafloor in the ocean domain is above −1600 m in most places and no deeper than -1730 m at present, vertical layers down to -2000 m were added to accommodate future retreat of the grounding lines into deeper terrain. The horizontal and vertical grid resolution were optimized to comply with the extent of the ocean domain, the parallel software architecture of MITgcm and the available computational resources on the UK HPC facility ARCHER2 ([www.archer2.ac.uk](www.archer2.ac.uk))...."*

*For Úa-MITgcm:*

*"The coupling timestep, or frequency at which data is exchanged between models, was set to 30 days. The short coupling interval was favoured to avoid the development of regularly spaced undulations in the ice draft of the Pine Island Ice Shelf, with kilometer-scale wavelength and amplitudes over 100 m.*

*The undulations are not a numerically robust feature of the coupled model, and a coupling timestep was chosen that suppresses their existence."*

5. *Reviewer 1 says:* "Last but not least, the conclusion could be a bit stronger in summing up the main findings – also to make sure that the authors' intentions are actually matched."

   *Reviewer 2 says:* "the findings in this section *(they refer to section 4 here)* are quite interesting, and it certainly could be useful to discuss their interpretation further in the conclusions."

   *We have rewritten the summary and conclusion section to better reflect the main findings of our work, as detailed in our reply to point 3 above. The new summary and conclusions are included in the manuscript with tracked changes at the end of this document.*

**Technical points raised by reviewer 1**

l. 15 „widespread dynamic thinning": Some ice research groups restrict the use of the term „dynamic thinning" to the meaning „thinning by flow field divergence". This does (to first order) not contribute to sea-level rise. I assume the authors use „dynamic" for „rapid, and with a lot of variability". Unless the authors do imply the strict meaning, I suggest to remove the word.

*We have removed "dynamic" from this sentence.*

l. 19: „imbalance" is a euphemism for „mass loss" here, isn't it? I suggest to be clear and use „mass loss"

*We have replace "imbalance" by "mass loss".*

l. 21: It should be either „For future decades to centuries," or the „in future decades to centuries" needs to be shifted behind „indicate that"

*We changed the sentence to "For future decades..."*

l. 26: no comma after „ice shelves"

*Done*

l. 41: no comma after „centuries"

*Done*

l. 46: no comma after „cavities"

*Done*

l. 85: no comma after „interface"

*Done*

l. 104: given the boundaries of the regional model, I wonder whether „10 vertical levels with 40 m

resolution down to 2000 m" actually exist (to that depth). Besides, the leves are horizontal, not vertical (it's the vertical coordinate, not vertical levels).

*We have adjusted the text to say "10 layers" instead of "10 vertical levels" and added an extra sentence to clarify our choice to add ocean layers to a depth of −2000 m:*

*"Although the seafloor in the ocean domain is above −1600 m in most places and no deeper than -1730 m at present, vertical layers down to -2000 m were added to accommodate future retreat of the grounding lines into deeper terrain. The deep basins of Kohler and Thwaites glaciers in particular, reach depths down to -2000m."*

l. 105: is the representation of ice and bottom topographies piecewise linear (=shaved cells) or piecewise constant = stepwise (=partial cells)?

*We thank the reviewer for spotting this mistake. We used a piecewise constant representation of geometry and have modified this in the manuscript.*

l. 114: Given the potential effects of deep convection on the continental shelf (outside the cavity), the assumption of „thermohaline properties in the deepest part of the cavities" to be „largely unaffected by changes in surface waters" seems daring. Isn't the (variability of) processes on the continental shelf one aspect of what coarse-resolution ice sheet models using melt rate parameterizations consistently tend to miss? I suggest to remove the sentence.

*Following this comment, and a similar comment by reviewer 2, we have removed the sentence from the manuscript.*

l. 156: please add a word or two on how the two time slices for hi_melt and lo_melt were chosen and how the respective data sets were created. Reasoning is clear and convincing, just what exactly has been done could be explained a bit more.

*We have expanded the text for the hi_melt experiment as follows:*

*"The choice of November 2002 conditions was solely based on their high melt bias in the histograms in Figure 2. Other selection criteria, such as the net heat flux into the cavities, were considered, but did not lead to a significantly different choice of ocean state or melt rates."*

*And for the av_melt experiment:*

*"Similar to the hi_melt experiment, the choice of January 1998 conditions was based on the histograms in Figure 2, with melt rates that are characteristic for the January 1997-December 2014 period. Other choices of boundary conditions might have led to a comparable ocean state and melt rates but were not tested."*

l. 165: inform _on_ the future evolution

*Done*

Section 3: stating that all variables are time-dependent and removing the „(t)" in all equations would improve readibility. If all variables were defined as also space-dependent except for averaged properties, which could be denoted with an overbar, the „(x)" could go too. Whereever possible (including in all the

text obviously), the use of K instead of °C is encouraged, because K is an SI unit, while °C is not.

*We have simplified the notation in section 3 to implicitly assume x and/or t dependency of all variables. However, we explicitly state if variables are time and/or space dependent when their notation is first introduced, and we use the overbar to denote spatially averaged quantities. We have replaced °C with K in the definition of all thermodynamic constants, such as the heat capacity, and the units of the momentum exchange coefficient, but continue to use units of °C to refer to actual temperatures (e.g. Table 2), in keeping with a long-standing convention in physical oceanography.*

Given that the T*s are temperature differences by unit and definition, the term „thermal driving" or „thermal forcing" may not be ideal. How about „melt forcing temperature" (with unit K again) or similar? The term „friction velocity" is widely used, despite the fact that it is not a „real" velocity, so there is a parallel. In any case, there should be ONE term defined and then consistently used and no switching between „thermal forcing" and „thermal driving".

*We opted to use the terminology "thermal driving" throughout the manuscript, given its widespread use in the literature, including the paper by Holland and Jenkins (1999), where the 3-equation formulation of basal melt was first introduced.*

*Holland, D. M. and Jenkins, A.: Modeling thermodynamic ice-ocean interactions at the base of an ice shelf, Journal of Physical Oceanography, 29, 1787–1800, https://doi.org/10.1175/1520-0485, 1999.*

caption to Figure 3: „oceanic boudary layer" may be more appropriate than „oceanic mixed layer" (because the important and correct-for-sure thing is that the layer is a boundary layer, while it being well-mixed may be something we can be less sure of)

*Whilst this is certainly true for the real ocean, in the model world we do not resolve the boundary layer, which is typically constrained to a thin region within 10s of centimeters to a few meters from the ice-ocean boundary. Since the discussion in Figure 3 refers to model results, we prefer to use the term 'mixed layer', which is more accurate in this case. It refers to 20m-thick grid cells next to the ice base, which are well mixed and have properties that are more representative for the far-field, i.e. well outside the physical boundary layer. We have added the following sentence to section 3.2 to clarify our choice of terminology:*

*"Cells adjacent to the ice-ocean interface will be referred to as the ``mixed layer'' to indicate that any vertical shear and thermohaline gradients that might occur in a narrow boundary layer adjacent to the ice base remain unresolved due to the 20 m vertical resolution of the model."*

Formal structuring of Section 4 is not ideal. It is recommended to add a subsection heading „4.1 Evolution of melt rates and cavity geometry" (or similar) right between lines 291 and 292. The following subsection headings need to be renumbered then, obviously.

*We followed the reviewer's advice and have added a new subsection heading, "4.1 Evolution of melt rates and cavity geometry". We have also renamed section 4 from "Ice-ocean feedbacks" to "Ice-ocean feedbacks in the absence of far-field ocean variability" to highlight the difference with section 5 on "The importance of far-field ocean variability".*

l. 292: no komma after „cavities"

*Done*

Table 2.: Some discussion of the physics behind the different values for the four coefficients seems desirable. Anything to be learned from the differences and consistencies between them? Also, how can μ become >1?

*The reviewer raises a good point about the physical meaning of μ > 1. We touched upon this point in line 344 ("At time t = 0, values of μ are between 1 and 1.18 (Table 2), which implies that the thermal forcing in the deep interior, $T_{DI}$ , is equal to or somewhat higher than $T_{IF}$"). To better clarify this point, we now write:*

*"At time t = 0, values of μ are between 1 and 1.18 (Table 2), which implies that on average, water masses that reach the deep cavity below –400m have an equal or slightly higher thermal driving compared to water masses that cross the ice front. The amplification of thermal driving can occur through the complex interplay between ocean dynamics and topography, resulting in the blocking of water masses with lower thermal driving, which are typically found at shallower depths."*

Caption to Figure 4: Is the GL color really magenta? Not red?

*The figure was plotted using rgb code [255,0,255] for magenta and looks magenta on our screen, so we kept the description in the caption.*

l .312: The „in turn" at the end of the sentence may be unnecessary.

*We have removed "in turn".*

l. 327: occurs

*Done*

l. 353: replace „ice draft" by „ice base"

*Done*

l. 369: suggest to replace „mixed layer" by „boundary layer"

*See our reply to the question about Figure 3 above.*

l. 373: same, also for l. 376, l. 386, and l. 442

*See our reply to the question about Figure 3 above.*

l. 374: delete „years"

*Done*

l. 391/392: If a transfer coeffficient (here epsilon_T) is determined by the forcing, does that not mean that the transfer coefficient is not well designed to characterize the transfer process it is supposed to describe? Please discuss. Also the mu>>1 calls for a bit of discussion. How can a transfer increase a signal?

*Regarding the first point, a transfer coefficient (or transfer function) by definition relates the input of a dynamical system to the output. In the case of the thermal transfer coefficient $\varepsilon_T$, the "input" is the thermal driving of the deep inflow, which indirectly depends on the far-field forcing through the outer cavity transfer coefficient, and the "output" is the thermal driving of the mixed layer. In this case, the transfer coefficient is simply defined as the ratio between output and input, i.e. the ratio between thermal driving of the mixed layer divided by the thermal driving of the inflow.*

*Regarding the second point, the statement in line 392 relates to \tilde{µ}, which is µ normalized by its value at t=0. Therefore, \tilde{µ} > 1 means that at time t>0, the value of µ is larger than at the start of the simulation. This was clarified a few paragraphs earlier:*

*"A time series of \tilde{µ}, which is the quantity of relevance for the calculation of the normalized melt rates in Eq. (13), is depicted in the third row of Figure 5. Values higher (lower) than one correspond to an increased (decreased) connectivity between the ice front and deep interior compared to the start of the simulations."*

*However, since µ(t=0) is larger than or equal to one according to Table 2, \tilde{µ}>1 also means that µ is always larger than one. The physical interpretation of µ > 1 was discussed above, following a comment about Table 2.*

l. 417/418: This is an interesting finding that could well re-appear in the Conclusions.

*We have reiterated this point in the conclusions:*

*"It was shown that changes in U$_*$ are dynamically linked to changes in the barotropic and overturning circulation, whilst no significant relationship was found between changes in melt and simple geometric characteristics, such as the gradient or depth of the ice base. The changes in melt are therefore controlled by complex adjustments of the ocean dynamics in response to the 3D evolution of the cavity geometry."*

l. 419-439: This is all plausible and interesting, but why is the x-y circulation adressed so nicely and the classical y-z ice-pump overturning circulation not at all?

*This is related to the next comment, so we provide a reply to both questions below.*

l. 463: same here: Why (only) look at the barotropic streamfunction and/but not at the overturning?

*We agree that the evolution of the barotropic and overturning flow are tightly connected. We have rewritten large parts of section 4.5 (formerly 4.4) to better reflect the concurrent changes in barotropic and overturning circulation, both from a general process point of view, and for each ice shelf individually. The manuscript with tracked changes at the end of this document contains the new version of section 4.5. In addition, we have simplified figure 7 by removing the timeseries of the depth-average flow, as it did not provide any new insights compared to the timeseries of the barotropic streamfunction amplitude in the same figure.*

l. 469: replace „modulate" by „dominate"?

*Done*

l. 476: This is one of the few places where the av_melt experiment appears. Is it needed at all for the point of this paper? (This is an open question, not a disguised suggestion.)

*The reviewer is right to point out that the main conclusions would not significantly change if the av_melt experiment were to be removed from the manuscript. However, this is an interesting observation in itself. Indeed, the physical processes that control melt-geometry feedbacks are shown to be independent of the specific realization of far-field ocean forcing used in the hi_melt and av_melt experiments. Only the timescales at which those feedbacks unfold, are dependent on the forcing. This is an important point to make, which can only be done when the av_melt experiment is included. We have reiterated this point in section 4.5, and in the summary and conclusion section.*

*"Whilst the discussion above was based on results from the hi_melt experiment, panel j in Figure 5 shows that an equally significant relationship between $\varepsilon_U$ (and hence $U_*$) and melt rates exists for the av_melt experiment. The dominant impact of geometrically induced changes in cavity circulation on the melt rates is therefore independent of the specific realization of far-field ocean forcing used in the hi_melt and av_melt experiments. The only key difference is the timescale at which those geometry-melt feedbacks unfold. Compared to the hi_melt experiment, the cavity geometries in the av_melt evolve more slowly due to the lower ocean thermal forcing, which causes an associated delay in the melt-rate response."*

*and*

*"The physical processes that control melt-geometry feedbacks operate independent of the specific realization of far-field ocean forcing. Only the timescales over which the feedbacks unfold dependent on the details of the forcing."*

l. 485/486 „recently" is actually with the Ice2sea project and the papers spawned from that. Which was in 2012, not 2022.

*We apologize for the lack of references to the existing literature. We have expanded the list to reflect the great progress that has been made to assess future ice-shelf melt rates, both in stand-alone ocean simulations and coupled ice-sheet-ocean simulations. We have also expanded the text, as follows:*

*"On the other hand, the impact of past and future anthropogenic climate change on ice-shelf melt in the Amundsen Sea has only recently started to be addressed in global (Timmermann and Hellmer, 2013; Naughten et al., 2018; Siahaan et al., 2022) and regional (Naughten et al., 2022; Jourdain et al., 2022a) ocean model setups with thermodynamically active ice shelf cavities."*

l. 486-488: The authors are obviously well aware that coupled ice sheet/ocean models with varying cavity geometry do exist, so their claim that „the interplay between […] far-field ocean variability and geometrically-driven changes in melt rates has not been included in numerical simulations" seems a bit odd. This needs to be modified to more precisely match what the authors intend to day.

*We thank the reviewer for pointing out our sloppy use of wording. To avoid confusion, we have rewritten the sentence as follows:*

*"Of the above studies, only Siahaan et al. (2022) included a dynamically evolving ice sheet. However, the 1° resolution of their global ocean configuration was too coarse to reliably capture the spatial distribution of melt rates beneath the small Amundsen Sea ice shelves."*

*instead of "In all cases though, the interplay between different modes of far-field ocean variability and geometrically-driven changes in melt rates has not been included in numerical simulations."*

l. 503: replace „orange" by „red"

*Done*

l. 503/504.: While the statement in „Results demonstrate that" is plausible, it is not actually demonstrated (in Fig. 8).

*We have replaced 'demonstrate' by 'suggest'.*

l. 524: Is the reference at the end of the sentence actually needed for the stamement made?

*The reference to (Jourdain et al., 2017) has been removed.*

l. 534: solid and dashed

*Done*

l. 536: „have a much lower amplitude" –> except for the jump for PIG, no?

*We agree that this sentence was misleading since we were referring to the hindcast experiment here, which only covers the first 17 years of the simulation. We have clarified this as follows:*

*"The corresponding variations in grounding line discharge (red line) and ice front flux (orange line) during the hindcast period have a much lower amplitude."*

l. 589/590: are the „sudden or gradual increases in grouding line discharge" created by enhanced ice flow or by grouding line migration?

*In general, changes in grounding line discharge are linked to both changes in grounding line location and changes in ice flow, as dictated by the complex non-linear and non-local relationships between grounding line flux, ice thickness at the grounding line and backstress at the grounding line. See e.g. Schoof (2007) for an analytical treatment in the simplified case of a flowline configuration. We did not analyze the details of the ice dynamics feedbacks here, which can be done more easily and comprehensively in a computationally cheaper stand-alone ice sheet model.*

*Schoof, C.: Ice sheet grounding line dynamics: Steady states, stability, and hysteresis, J. Geophys. Res.-Ea. Surf., 112, F03S28, https://doi.org/10.1029/2006JF000664, 2007a*

l. 609: The phrase „[…] offsetting shifts in grounding line discharge" needs to be clarified

*We have rewritten the conclusions, and this sentence was removed in the process.*

l. 613: replace „represented" by „as indicated" ?

*Done*

**Technical points reviewer 2**

Minor typographical errors are present and there are a few instances of complex sentences that could be simplified for clarity.

**Line 21-24**: "In future decades to centuries, numerical mass balance projections indicate…." -Consider restructuring the sentence (or splitting it into two) for clarity.  For example, "In future decades to centuries, numerical mass balance projections indicate that the Amundsen basin is likely to remain Antarctica's dominant contributor to sea level rise. This persists despite significant uncertainties in climate forcing and poorly represented physical processes, such as ice-shelf melting and temporal changes in ice rheology, basal sliding, and ice front location."

*We have split the sentence in two, as suggested.*

**Line 35-38**: "In recent decades ice-shelf thinning rates have decreased..." -Consider restructuring the sentence for clarity.

*We have restructured the sentence and added a reference to a recent publication by Paolo et al. (2023), who address exactly this point:*

*"In recent decades, ice-shelf thinning rates in the Amundsen Sea have decreased (Adusumilli et al., 2018) despite an up to two-fold increase in grounding line flux (Mouginot et al., 2014; Davison et al., 2023; Otosaka et al., 2023). A combination of processes that could have led to a reduction in basal melting, such as shoaling of the ice-shelf base and deepening of the thermocline depth on the continental shelf, has been suggested as a potential cause (Paolo et al., 2023). However, validating the delicate interplay between different ice-shelf mass balance components in the absence of a long-term and reliable record of ice surface velocities, ice-atmosphere interactions and ocean hydrography remains a difficult problem. Hypothetically, a continued decrease in ice-shelf thinning could drive the system towards a new steady state, although there is no evidence from numerical simulations that such a steady state can be obtained under present-day ocean and atmospheric conditions (Arthern and Williams, 2017; Reese et al., 2023, e.g.)."*

**Line 41**: "The rate at which glaciers in the Amundsen basin will continue to lose mass over the next decades to centuries, is controlled " -the comma after "centuries" is unnecessary.

*We removed the comma.*

**Line 46**: "A third, as-of-yet poorly understood process, is the potential feedback between changes in the geometry of ice-shelf cavities, and the ocean dynamics …." – the comma after "cavities" is unnecessary.

*We removed the comma.*

**Line 58**: "(Edwards et al., 2021, e.g.)," -the use of "e.g." seems unnecessary.

*We removed "e.g." here*

**Line 87**: "(Patmore et al., 2019, e.g.)" - the use of "e.g." seems unnecessary.

*We removed "e.g." here*

**Line 114-116:** -Consider assessing and re-writing the entire sentence "thermohaline properties in the deepest parts of the cavities, which are thought to be largely unaffected by changes in surface waters." The decision to ignore surface fluxes is a valid approach in order to focus mainly on the interactions between cavity geometry and basal melt. However, the statement "thermohaline properties in the deepest parts of the cavities, (which) are thought to be largely unaffected by changes in surface waters" doesn't seem correct.

*In reply to this comment and a similar comment by reviewer 1, we have removed this sentence from the manuscript.*

**Line 326-327:** "High-frequency fluctuations at monthly timescales are predominantly caused by Eddy activity at the ice front, which occur irrespective of the changes in cavity geometry..." –adjust the verb to "occurs" if the sentence is grammatically consistent.

*Done*

**Line 369-370:** Please explain the meaning of "changes in the mixed layer speed" as I might have misunderstood the usage here.

*We have clarified what we mean by 'mixed layer speed':*

*"However, it will be shown in Sect. 4.5 that despite the increase in ambient ocean thermal driving, melt rates of the Dotson Ice Shelf still decrease overall due to a reduction in the friction velocity relative to the ice base ($U_*$ in Eq. (2))"*

**Line 404-405:** "The fourth and final coefficient to control average melt rates..." -The phrase might be more grammatically correct as "The fourth and final coefficient controlling average melt rates..."

*Done*

**Line 419:** "Whilst basal gradients do locally impact on the buoyancy of the mixed layer..." -Consider changing "impact on" to "impact".

*We replace "impact on" by "affect"*

**Line 480 and 492:** The phrase "geometrically driven" could be hyphenated as "geometrically-driven" for clarity.

*Done*

**Line 489:** Small typo in "Embayement" -Consider changing it also to Amundsen Sea Embayment.

*Done*

**Line 494-495:** Consider changing "a" to "an" in "over a 18-year period"

*Done*

**Line 534:** It should be "and" in "(solid an dashed blue lines in Fig. 9)"

*Done*

[revised manuscript text omitted]